# Structure-based design of a phosphotyrosine-masked covalent ligand targeting the E3 ligase SOCS2

Sarath Ramachandran [1,4], Nikolai Makukhin[1,3,4], Kevin Haubrich [1], Manjula Nagala[1], Beth Forrester[1], Dylan M. Lynch[1], Ryan Casement[1], Andrea Testa[1,3], Elvira Bruno[1], Rosaria Gitto[2] & Alessio Ciulli [1]✉

The Src homology 2 (SH2) domain recognizes phosphotyrosine (pY) post translational modifications in partner proteins to trigger downstream signaling. Drug discovery efforts targeting the SH2 domains have long been stymied by the poor drug-like properties of phosphate and its mimetics. Here, we use structure-based design to target the SH2 domain of the E3 ligase suppressor of cytokine signaling 2 (SOCS2). Starting from the highly ligand-efficient pY amino acid, a fragment growing approach reveals covalent modification of Cys111 in a co-crystal structure, which we leverage to rationally design a cysteine-directed electrophilic covalent inhibitor MN551. We report the prodrug MN714 containing a pivaloyloxymethyl (POM) protecting group and evidence its cell permeability and capping group unmasking using cellular target engagement and in-cell [19]F NMR spectroscopy. Covalent engagement at Cys111 competitively blocks recruitment of cellular SOCS2 protein to its native substrate. The qualified inhibitors of SOCS2 could find attractive applications as chemical probes to understand the biology of SOCS2 and its CRL5 complex, and as E3 ligase handles in proteolysis targeting chimera (PROTACs) to induce targeted protein degradation.

Protein–protein interactions (PPIs) play important roles in regulating cellular processes including enzyme catalysis, cell signaling and development, and protein homeostasis. For these reasons, targeting PPIs, directly or allosterically, provide compelling strategies for modulating protein function. However, it remains challenging to design and develop small molecules that bind with high affinity and specificity to PPI sites. The Src homology 2 (SH2) domain, found in over 110 human proteins, serve as a key mediator of PPIs facilitating recognition and binding of phosphorylated tyrosine residues in partner proteins[1]. SH2 domain-containing proteins are attractive therapeutic targets due to dysregulation in many diseases, including cancer[2]. Significant efforts to target SH2 domains in the 1990s and early 2000s largely proved unsuccessful[3,4], and as a result SH2 domains have been dubbed as undruggable. A major challenge associated with targeting SH2 domains is their highly polar PPI surface and in particular the pocket that binds to phosphotyrosine (pY)[4]. Peptide-based ligands containing pY, despite their high affinity, suffer from poor drug-like properties such as low cell permeability, susceptibility to proteolytic cleavage, and enzymatic lability of the phosphate group. Recent advances in the development of phosphate group analogs (phosphomimetics) chemistries and phosphate-masking capping prodrugs have offered alternative strategies, however, have largely remained niche and mostly target-dependent in scope[1,5–9].

[1]Centre for Targeted Protein Degradation, Division of Biological Chemistry and Drug Discovery, School of Life Sciences, University of Dundee, 1 James Lindsay Place, Dundee DD1 5JJ, United Kingdom. [2]Department of Chemical, Biological, Pharmaceutical, and Environmental Sciences, University of Messina, Viale Stagno D'Alcontres 31Pole Papardo 98166 Messina, Italy. [3]Present address: Amphista Therapeutics Ltd, Cory Building, Granta Park, Great Abington, Cambridge CB21 6GQ, United Kingdom. [4]These authors contributed equally: Sarath Ramachandran, Nikolai Makukhin. ✉e-mail: a.ciulli@dundee.ac.uk

An attractive class of targets containing SH2 domains that have remained unliganded to date is the suppressor of cytokine signaling (SOCS) family of proteins. SOCS proteins can modulate cytokine signaling pathways by inhibiting the Janus kinases (JAKs) directly. In addition, because many SOCS proteins can form part of Cullin 5 E3 ligase complexes, SOCS proteins can inhibit the JAK/STAT pathway indirectly by targeting signaling components such as the cytokine receptors for ubiquitination and subsequent proteasomal degradation[10,11]. Eight family members are known, SOCS1-7 and cytokine-inducible SH2 containing protein (CISH), that share a conserved domain architecture with the SH2 domain functioning as the substrate adaptor mediating binding to phosphorylated partner proteins. We have been particularly interested in SOCS2 because of its role in suppressing signaling by a variety of stimuli, including growth hormone, erythropoietin, prolactin, and interleukin, and due to its links to disorders of the immune system, central nervous system, and cancer[12,13]. In addition to its suppressive role, SOCS2 is unique within the SOCS protein family in its capability to potentiate rather than suppress signaling, by antagonizing SOCS1 and SOCS3 activity[14]. SOCS2 can tightly bind to adaptor subunits ElonginB/C and scaffold subunit Cullin 5 and RING domain protein Rbx2, to function as substrate receptor of Cullin 5 RING E3 ligase complex CRL5^SOCS2. Ubiquitin E3 ligases drive the selectivity of the ubiquitin-proteasome system (UPS) by recognizing and targeting substrate proteins for ubiquitination[15]. Drug discovery efforts targeting E3 ligases thus offer a selective mode of therapeutic intervention[16]. A number of small molecules targeting the substrate adaptors of Cullin RING E3 ligases have been discovered, notably for the CRL2 von Hippel-Lindau (VHL) protein, for the CRL4 cereblon (CRBN) and DCAF15, and for the CRL3 KEAP1[17–22]. These compounds have utility as chemical probes, either as inhibitors of the specific E3-substrate interaction to block substrate ubiquitination or degradation, or molecular glues to stabilize recruitment of neo-substrate proteins, such as the case of thalidomide and other immunomodulatory drugs with CRBN, and aryl-sulfonamides drugs with DCAF15. In addition, such E3 ligase ligands can be successfully incorporated into proteolysis targeting chimeric molecules (PROTACs) to induce degradation of proteins of interest, with the most popularly used being the VHL and CRBN ligands. The limited numbers of small-molecule E3 ligands being efficiently used for PROTACs have motivated efforts to expand the E3 handle toolbox to diversify the hijacked E3 biology, opening opportunities to enable tissue/disease-specific targeting and addressing resistance mechanisms currently emerging with VHL and CRBN PROTACs[23,24].

Previous work from us and others solved crystal structures of the unbound SOCS2-ElonginB-ElonginC (SBC2) complex and characterized low nanomolar binding affinities between SBC2 and Cullin 5, as measured using isothermal titration calorimetry (ITC) and surface plasmon resonance (SPR)[25–27]. We later showed using pulldown experiments with phosphorylated growth hormone receptor (GHR) peptides that the bound SOCS2 protein eluted together with all the expected subunits part of the CRL5 ligase complex, and we fully characterized the structural assembly of the CRL5^SOCS2 complex in solution[28]. More recently, our laboratory revealed structural insights into the SOCS2-substrates interactions by solving co-crystal structures of SBC2 in complex with phosphorylated peptides from substrates growth hormone receptor (GHR pY595) and erythropoietin receptor (EpoR-pY426)[29]. The structures revealed canonical SH2 domain–substrate interactions around the key phosphotyrosine (pY) residue, and mainly backbone-directed interactions with the residues surrounding pY. The phosphate group of pY of substrate peptides is crucial for SOCS2 binding, as exemplified by the binding affinity of GHR pY595 peptide ($K_D = 1.1\,\mu M$) that is completely abrogated with a non-phosphorylated analog GHR Y595[28]. Co-crystal structures revealed that the phosphate group forms key interactions with the SH2 domain of SOCS2, via a network of direct and water-mediated hydrogen bonds[29]. Despite the potential therapeutic utility of ligands targeting the SOCS proteins, efforts have largely been limited by the aforementioned challenges involved with targeting the SH2 domain.

In this work, we describe the structure-guided design, using fragment-based growing, of a pY-based covalent small-molecule binder targeting the SOCS2-SH2 domain. A co-crystal structure with one of our high-affinity analogs serendipitously identifies modification of Cys111 on SOCS2, which we later rationally exploit via chloroacetamide electrophilic substitution, yielding covalent inhibitor **MN551**. Site-selectivity and efficiency of covalent **MN551** is evidenced by co-crystal structures and further characterized in vitro using both intact electrospray ionization mass spectrometry and a fluorescence polarization assay. To circumvent the lack of cell permeability of the naked pY group, we take a prodrug approach to mask the negative charge of the phosphate group. We develop a split-NanoLuc-based assay and apply it to demonstrate cellular target engagement of SOCS2 and to identify the pivaloyloxymethyl (POM) moiety as the best prodrug strategy. We show rapid prodrug unmasking of our POM-protected inhibitor **MN714** using in-cell $^{19}$F-NMR spectroscopy. We further evidence that our SOCS2 inhibitor competitively blocks recruitment of cellular SOCS2 protein to its native substrates via covalently modifying Cys111 using peptide-pulldown and mass spectrometry proteomics.

## Results

### Structure-guided design and optimization of non-covalent SOCS2 binding ligands

Armed with learnings and successes from our previous structure-guided design campaign to develop VHL ligands starting from hydroxyproline as the PTM-recognition unit, we hypothesized that small-molecule SOCS2 ligands could be rationally designed using phosphotyrosine as a suitable starting point anchor fragment for binding to the SH2 domain and disrupt the SOCS2-substrate interaction. Phosphotyrosine and its capped analog **1** bound to SOCS2 with surprisingly high binding affinity, each with $K_D = 190\,\mu M$, respectively, as measured by isothermal titration calorimetry (ITC)[28,30]. While this corresponded to 100-fold weaker binding than the full phosphorylated peptide, the small fragment-like size of these ligands corresponds to a ligand efficiency LE of 0.29 kcal/mol/NHA, which is excellent for targeting a PPI binding site, and so was deemed attractive to elect pY as the anchor fragment starting point. To increase the affinity, we sought to build the molecule from both the N- and C-terminal end of the pY fragment to explore interactions around the pY pocket of the SH2 domain.

First, we decided to optimize the N-terminal region of **1** while maintaining the N-methylamide group at the C-terminus. To generate analogs rapidly, we employed a solid phase synthesis using methyl indole AM resin. After Fmoc deprotection, the resin was loaded with Fmoc-Tyr (PO(OBzl)$_2$)-OH[29]. Subsequent coupling with various commercially available carboxylic acids and cleavage with TFA allowed the rapid synthesis of N-methylcarboxamide phosphotyrosines (Supplementary Fig. 1). We introduced different substituents at the N-terminus to cover a range of physicochemical properties (e.g. alicyclic, heterocyclic, aromatic with electron-withdrawing and electron-donating groups) to maximize potential interactions (see Table 1 for representative examples). Binding affinities of the synthesized compounds for SOCS2 were measured via two orthogonal biophysical binding assays: (1) surface plasmon resonance (SPR) with SBC2 immobilized on the chip (Supplementary Fig. 7); and (2) $^{19}$F ligand-observed displacement NMR assays, providing two orthogonal biophysical measurements of dissociation constants ($K_D$)[29,31]. We found an excellent correlation between the $K_D$ values measured via the two techniques (Table 1). Para-substituted benzyl moieties yielded the greatest contributions to binding affinity and there were only slight differences between electron-withdrawing groups such as a methyl ester and electron-donating such as

**Table 1 | Dissociation constant ($K_D$) values measurements for N-methylcarboxamide phosphotyrosines binding to SBC2 (n = 1)**

| Compd. | R | ¹⁹F NMR $K_D$, µM (LE, kcal mol⁻¹ NHA⁻¹)[a] | SPR $K_D$, µM (LE, kcal mol⁻¹ NHA⁻¹)[a] |
|---|---|---|---|
| 1 | Me | 186 (0.24) | 269 (0.23) |
| 2 | (cyclopropylmethyl) | 106 (0.22) | 126 (0.22) |
| 3 | (morpholinoethyl) | 352 (0.17) | 205 (0.18) |
| 4 | (benzyl) | 62 (0.21) | 114 (0.20) |
| 5 | (4-fluorobenzyl) | 32 (0.215) | 49 (0.21) |
| 6 | (4-methoxybenzyl) | 50 (0.20) | 58 (0.20) |
| 7 | (4-methoxycarbonylbenzyl) | 34 (0.19) | 44 (0.19) |
| 8 | (pyridinylmethyl) | 85 (0.20) | 75 (0.20) |

[a]LE values are shown for each compound as LE = −RT·ln(Kd)/NHA, where NHA is the number of non-hydrogen atoms[68].

methoxy group. At the same time, the 4-fluorobenzyl substituted ligand **5** was found to be the strongest binder in the series with comparable $K_D$ = 32 µM by ¹⁹F NMR and 49 µM by SPR, and good LE = 0.21 kcal/mol/NHA.

Having established the (4-fluorophenyl) acetamido group as the preferred substituent at the N-terminus, we sought to optimize the C-terminal fragment while fixing the N-terminus of the ligand (Fig. 1g). Close inspection of our substrate-bound co-crystal structures revealed a SOCS2-SH2 groove between Ile110 and Thr93 (named the "IT" channel) comprising of hydrophobic residues Leu95, Leu106, Leu150 and EF loop residues (Ile109, Ile110, Val112, and Leu116), that is utilized by both EpoR and GHR peptides to form hydrophobic interactions[29] (Fig. 1a, c). We hypothesized that SOCS2 ligands could be optimally grown out of the C-terminus, by fitting snugly within the narrow groove of the IT channel, with a goal to access this largely hydrophobic region. Moreover, the flexible nature of the flanking EF loop (residues 107-116) can help accommodate a wide range of chemical groups in the groove. Towards this aim, we designed and synthesized a next library of SOCS2 ligands using commercial available Fmoc-Tyr(PO(NMe₂)₂)-OH as a starting material. Fmoc-Tyr(PO(NMe₂)₂)-OH was coupled with various benzylamines followed by Fmoc deprotection and subsequent coupling with 4-fluorophenylacetic acid. Deprotection of the N,N-dimethyldiamide phosphate-protecting group with TFA afforded the desired phosphotyrosines (Supplementary Fig. 2). The binding of all obtained molecules was quantified using ITC and SPR (Table 2). This

second-round of structure-activity relationship (SAR) exploration identified several ligands with improved binding affinity. We found that 4-fluorobenzylamino **9** and 3-methylbenzylamino **10** improved binding affinity by 10-fold compared to **5**, showing Kd (SPR) = 5 µM and 3 µM, respectively (Fig. 1g, Table 2). ITC titrations yielded Kd of 2.6 µM for **9** and 1.1 µM for **10**, comparable to SPR, and evidenced exothermic binding ΔH = −4.1 and −5.8 kcal/mol (Table 2, and Supplementary Fig. 8).

To elucidate the ligand binding mode, we solved the crystal structures of SBC2 in complex with **9**, via ligand soaking (Fig. 1b). Many of the key interactions formed by the pY residue are conserved between the ligand and the substrate peptides[29]. These include: (1) hydrogen bonds between the phosphate group of the ligand and SOCS2-SH2 pocket residues Arg73, Ser75, Ser76, and Arg96; (2) cation-π interaction between the pY aromatic ring and Arg96; and 3) hydrogen bonds between the amide bonds surrounding pY and the backbone amides and side chain groups of Thr93 and Asn94. Pleasingly, the C-terminal 4-fluorobenzylamino group of **9** successfully occupied and engaged the hydrophobic patch residues Thr93, Leu95, Ile110, and Leu150 of the IT channel, as designed (Fig. 1b).

## Serendipitous discovery of covalent modification of SOCS2 Cys111

With the structure of **9** in hand, further optimization attempts focused towards engaging an attractive hydrophobic groove identified

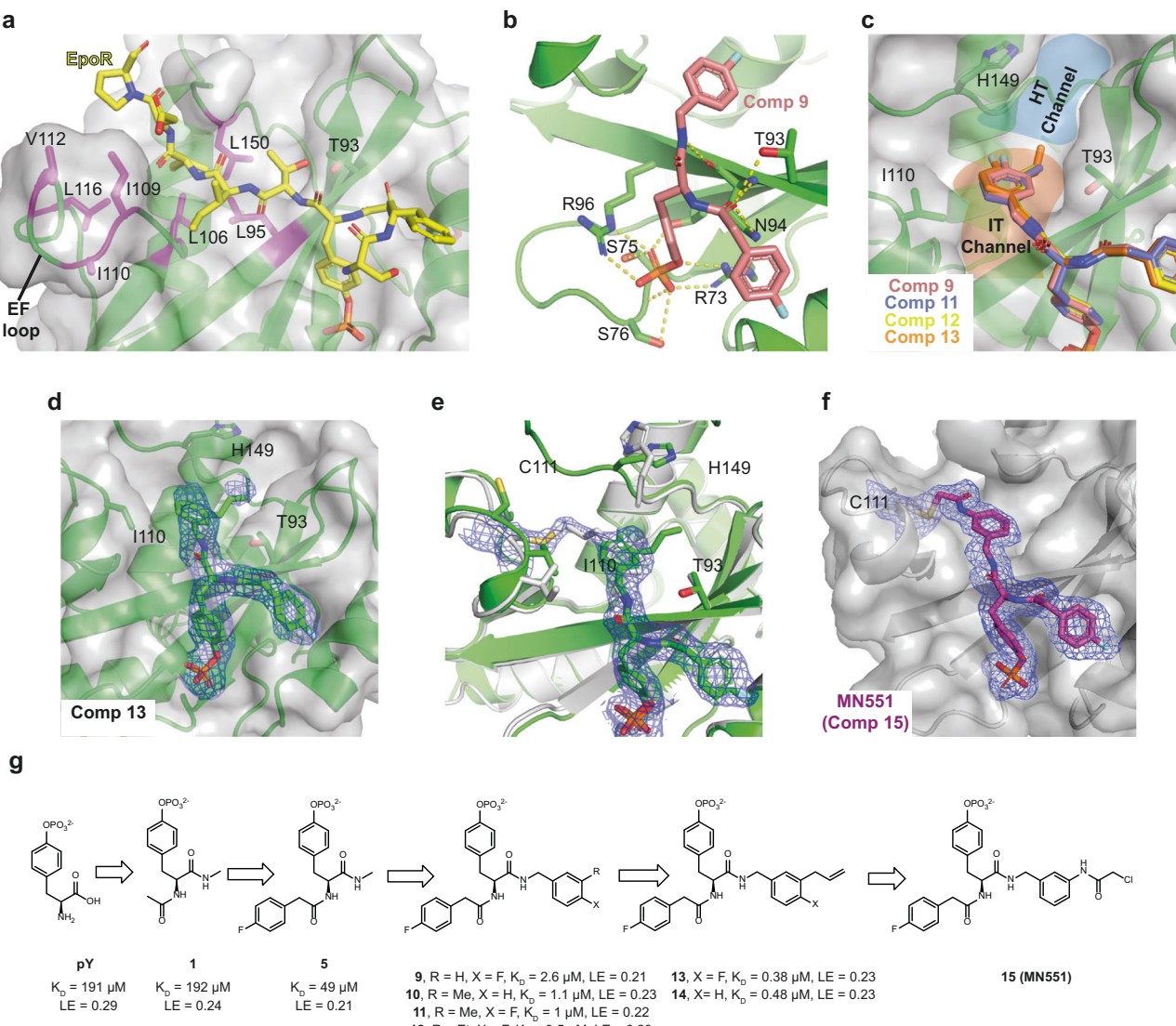

**Fig. 1 | Rational crystallography-guided design of small-molecule ligands targeting the SH2 domain of SOCS2. a** Crystal structure of SOCS2 in complex with bound EpoR peptide (yellow carbons, PDB: 6I4X). SOCS2 residues participating in hydrophobic interactions with the substrate peptide are highlighted in magenta. **b** Crystal structure of the binary complex SBC2-**9**. Hydrogen bonds formed between compound **9** (pink carbons) and SOCS2 residues (green carbons) are shown as yellow dash. **c** Superposition of co-crystal structures of SOCS2 with bound compounds **9** (pink carbons), **11** (blue), **12** (yellow), and **13** (orange) evidence substantial overlap in binding mode of the benzylic group at the protein's IT channel (highlighted in orange) and the HT channel (blue) pockets targeted for ligand design. **d** SBC2 crystals soaked with compound **13**. SA omit map (blue mesh) is shown contoured at 3σ. **e** superimposition of soaked (green carbons) and co-crystallized (light gray) SBC2-**13** structures highlighting the major differences in ligand conformation. Blue mesh represents the 2Fo-Fc map contoured at 1σ for the co-crystallized **13** structure. **f** Structure of compound **15** (**MN551**) co-crystallized with SBC2 evidencing clear covalent bond formed with Cys111 from the EF loop of SOCS2. SA omit map for the ligand displayed as blue mesh and contoured at 3σ. **g** Fragment-based design of small-molecule SOCS2 ligands by stepwise growing around the anchor core fragment pY. $K_D$ values for all compounds, except **5**, are reported from direct ITC experiments ($n = 1$, see full titration in Supplementary Fig. 8). LE values for each compound are reported in kcal/mol/NHA and calculated as LE = −RT·ln($K$d)/NHA, there NHA is the number of non-hydrogen atoms[68].

between His149 and Thr93 (named "HT" channel) (Fig. 1c). It was hypothesized that this channel could be readily accessible by further modifications of the 4-fluorobenzylamino group, particularly meta-substitutions. We therefore next synthesized compounds **11**, **12**, **13**, **14** by introducing methyl, ethyl or allyl group at position 3 of the benzylamine to form favorable hydrophobic interactions with the HT channel. Indeed, ligand **13**, bearing an allyl group, yielded the highest binding affinity (ITC $K_D$ = 380 nM, ΔH = −7.1 kcal/mol, Table 2 and Supplementary Fig. 8). To reveal the ligand binding modes, we solved crystal structures of SBC2 soaked with **11**, **12** and **13**. Surprisingly, upon inspection of the structures we realized that the electron density for the allyl group of **13** was largely missing in the bound ligand. We postulated that this observation could be a result of steric clash forcing

an unnatural binding mode, potentially resulting from pre-existing crystal contact constraints of apo SBC2 crystals preventing the soaked ligand to adopt its preferred binding mode.

To test this hypothesis and aim to capture more bona fide binding interactions for **13**, we decided to pursue a co-crystallization strategy next. SBC2 was co-crystallized in the presence of **13** and we successfully solved the structure (Fig. 1e). The liganded co-crystal structure had a different space group from the apo structure, which comprised of four protomers in the asymmetric unit in comparison to a single protomer for the apo crystal form. The ligand was bound in all four subunits, and in one of the protomers, we observed unusual electron density towards the opposite direction from the HT channel. Closer inspection of the omit map electron density in this region identified a

**Table 2 | Biophysical characterization of the second-round library of SOCS2 ligands (n = 1)**

| Compd. | SPR $K_D$ (µM) | ITC | | | | LE (kcal mol$^{-1}$ NHA$^{-1}$) |
|---|---|---|---|---|---|---|
| | | $K_D$ (µM) | ΔG (kcal/mol) | ΔH (kcal/mol) | −TΔS (kcal/mol) | |
| 9 | 4.2 | 2.6 | −7.65 | −4.1 | −3.55 | 0.21 |
| 10 | 3.7 | 1.1 | −8.12 | −5.81 | −2.31 | 0.23 |
| 11 | 2.7 | 1.0 | −8.23 | −5.25 | −2.98 | 0.22 |
| 12 | 2.2 | 0.51 | −8.59 | −8.16 | −0.43 | 0.23 |
| 13 | 2.4 | 0.38 | −8.75 | −7.13 | −1.62 | 0.23 |
| 14 | 2.4 | 0.48 | −8.61 | −6.31 | −2.26 | 0.23 |

covalent bond between the terminal carbon of the allyl group of the ligand and Cys111 at the EF loop (Fig. 1d, e). This observation was surprising and unexpected because allyl groups are not considered strong electrophiles and are not known nor used to readily react with cysteine residues, unless the thiol-allyl reaction is radically induced[32]. Nonetheless, this serendipitous finding suggested that Cys111, which is in the flexible EF loop of the SOCS2-SH2 domain, could potentially be targeted as a reactive nucleophilic protein residue. Curiously, our previous crystallographic work had identified Cys111 as highly reactive site to electrophilic modifications with cacodylate from the crystallographic buffer[26].

### MN551 is a Cys111-specific covalent SOCS2 ligand that blocks substrate binding

The observation that Cys111 could act as reactive nucleophilic residue on SOCS2 prompted us to design covalent binder 15 (hence afterward referred to as **MN551**), by replacing the allyl group in 14 with a more electrophilic chloroacetamido group. We chose 14 as scaffold ligand instead of 13 (hydrogen instead of fluorine at the para position) due to convenient availability of 3-(Boc-amino) benzylamine as reagent for the incorporation of the chloroacetamido group during the synthesis of **MN551** (Supplementary Fig. 3). The chloroacetamide moiety newly introduced in **MN551** covalently modified Cys111, as confirmed by protein X-ray crystallography and mass spectrometry experiments (Figs. 1f, 2a). A co-crystal structure of SBC2 preincubated with **MN551** for 2 h prior to crystallization revealed the formation of a covalent bond between the chloroacetamide group of **MN551** and Cys111. The binding mode for the remaining portion of the **MN551** molecule remained unchanged compared to the parental analogs. Only a 1:1 covalent SOCS2:**MN551** adduct was observed using intact mass analysis by electrospray protein mass spectrometry (Fig. 2a, b), and no reactivity was observed with a C111S mutant (Fig. 2c), consistent with high specificity of binding and modification site. Time-dependent reaction monitored by mass spectrometry showed that **MN551** covalently modified recombinant SOCS2 protein to stoichiometric occupancy within 2 h at Cys111 (Fig. 2c). Binding of **MN551** to SBC2 was further validated with DSF and ITC (Fig. 2d, e). **MN551** enhanced the thermal stability of SBC2 as demonstrated by a 6 °C shift in the melting temperature. The reversible binding affinity of **MN551** to SBC2 was determined by ITC to be $K_i$ = 2.2 µM (Fig. 2e). To confirm that **MN551** blocks binding of the natural substrates of SOCS2, we performed competition ITC experiments of GHR_pY595 peptide against SBC2 either unmodified or preincubated with equimolar amount of **MN551** for 2 h at room temperature. In the presence of **MN551**, GHR was no longer able to bind to SBC2, nor to compete out the inhibitor, demonstrating covalent saturation of the SOCS2 binding site by **MN551** (Fig. 2f).

With the specific covalent modification of Cys111 established, we decided to fully characterize **MN551** as a covalent SOCS2 inhibitor. To this end, we developed a covalent fluorescence polarization assay to

monitor displacement of a non-covalent fluorescent probe 26 ($K_D$ = 77.56 nM, as measured with FP, Supplementary Fig. 9) by **MN551**, to ultimately determine the covalent efficiency $k_{inact}/K_I$ that is a critical parameter to characterize covalent inhibitors[33] (Fig. 2g, h, Supplementary Fig. 9). Due to the covalent nature of the competing inhibitor, a time-dependent reduction in the fraction of protein-bound probe is observed. Plotting the rate ($k_{obs}$) of time-dependent reduction in probe FP signal with **MN551** concentrations, yields the parameters $k_{inact}$ = 2.1 × 10$^{-4}$ s$^{-1}$ (maximum potential rate of inactivation) and $K_I$ = 3.6 µM (concentration of **MN551** at which $k_{obs}$ = $k_{inact}$/2) (details in methods and Supplementary Fig. 9)[34]. We also calculated $K_i$ = 1.1 µM at time = 0 and unlike the $K_i$ determined by ITC measurement, calculated $K_i$ from the FP represents the initial reversible binding affinity of **MN551**. Figure 2h captures the initial rates to give an accurate estimation of the efficiency of covalent bond formation, $k_{inact}/K_I$ = 58 M$^{-1}$ s$^{-1}$. This value illustrates reasonable specificity for a covalent inhibitor, with nonetheless scope for future optimization. In addition to characterizing the kinetic covalency parameters, we have also assessed the stability of the **MN551** by monitoring its reactivity (half-life >70 min) with reduced glutathione (GSH) (Supplementary Fig. 10)[35].

### MN551 selectively reacts with SOCS2 and CISH over other SOCS family members

Having established the specificity of **MN551** for covalent modification of Cys111 of SOCS2, we turned our attention to the intra-SOCS family selectivity of **MN551**. We performed a multiple sequence alignment of the SH2 domains from all the SOCS proteins (Fig. 3a). The Cys111 targeted by **MN551** for covalency is part of the EF loop in SOCS2. Beyond SOCS2, other family members SOCS4, SOCS5, SOCS7, and CISH also have a cysteine in the same loop (Fig. 3a), and given the expected conserved ligand binding mode, they could all be engaged by **MN551**. To verify our hypothesis, we incubated recombinant SOCS/EloBC complexes of SOCS4 (SBC4), SOCS6 (SBC6), and CISH with **MN551** for 2 h and analyzed for covalent modification of the SOCS proteins by intact ESI-MS (Fig. 3b–d). These proteins were chosen as SOCS4 and SOCS6 are already structurally well characterized, and because SOCS4 Cys350, and CISH Cys144 are in a similar position as Cys111 in SOCS2, whereas SOCS6 is lacking cysteines in the EF loop[36–38]. We observed 18% modification of SOCS6 and a complete modification of CISH. In contrast, **MN551** failed to covalently modify SOCS4. An overlap of crystal structures of SOCS2 and SOCS4 shows that although the Cys350 from SOCS4 is in the same loop as Cys111 from SOCS2, the **MN551** binding mode and distance from the reactive group of **MN551** may not be ideal for covalent bond formation (Fig. 3e). Although SOCS6 lacks a cysteine in its EF loop, Cys471 from the BG loop is predicted to be in close proximity to the chloroacetamide functional group of **MN551**, as evident from the superposed SOCS2 and SOCS6 structures, potentially explaining the modification, albeit minimal (Fig. 3f). Structure-guided improvements upon this observed **MN551**-SOCS6 covalency could expedite future attempts in development of covalent binders for SOCS6. In the absence of a PDB structure for CISH, we overlayed SOCS2-SH2 domain with an AlphaFold model generated for the CISH SH2 domain with deletions in the N-terminus and the PEST sequence (66-258, Δ174-202, Fig. 3g). Amongst all the SOCSs, CISH has the highest sequence identity to SOCS2-SH2 and the presence of a conserved Cys144 in the EF loop together likely explain its complete covalent modification by **MN551**.

### POM-protected prodrug MN714 enables cell permeability and intracellular target engagement

One of the challenges with small-molecule binders displaying on-target activity with recombinant proteins, is to translate their utility to a cellular context. To demonstrate cell permeability and targeted engagement with our molecules, we developed a split-NanoLuc based

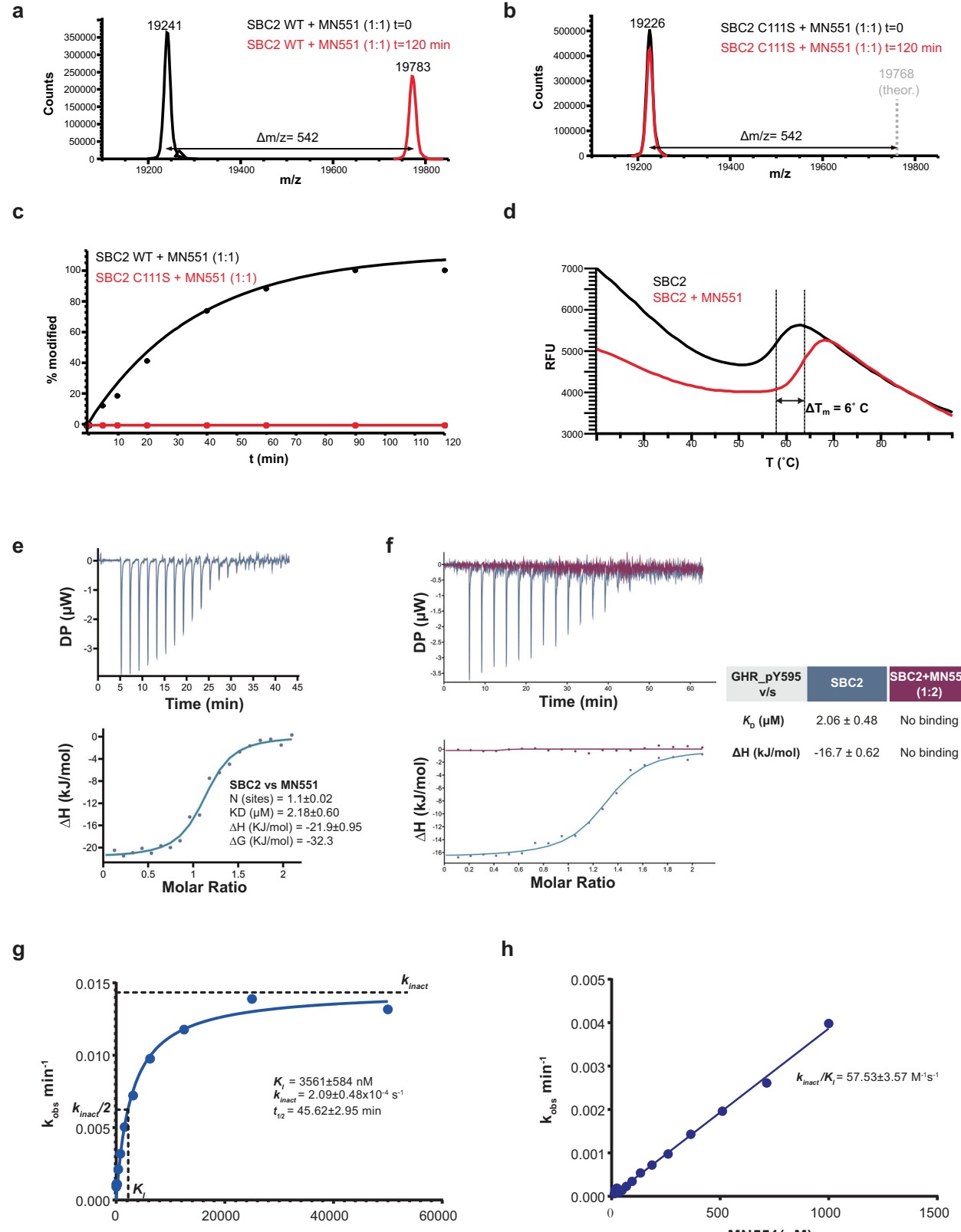

**Fig. 2 | In vitro characterization of MN551 covalency with recombinant SOCS2.**
**a** Mass-spec analysis on SBC (40 µM) treated with **MN551** (40 µM) reveals a single covalent adduct of **MN551** on SOCS2. **b** SOCS2 C111S mutant does not covalently react with **MN551**. **c** Time-dependent modification of SOCS2 by **MN551**. **d** DSF melting curves for SBC2 and SBC2:MN551 complex demonstrating stabilization of SBC2 with **MN551**. ITC measurements ($n = 1$) of **e** MN551 binding to the SBC2 **f** GHR_pY595 peptide binding to SBC2 or the preincubated complex of SBC2:MN551 in molar ratio of 1:2. **g**, **h** Characterization of kinetic parameters−$K_I$, $k_{inact}$, and $k_{inact}/K_I$ using FP assay (mean ± SEM, $n = 3$).

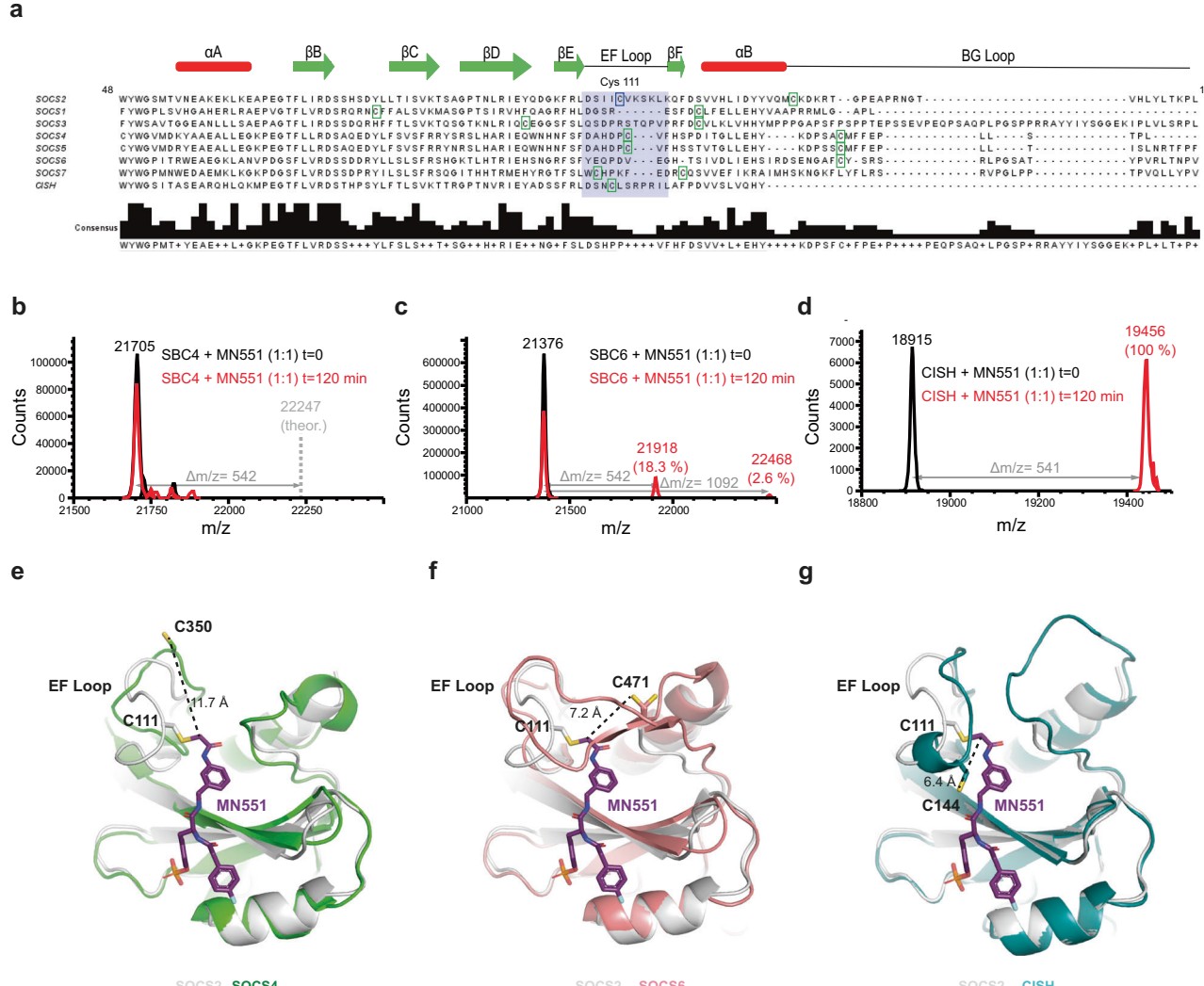

**Fig. 3 | Intra-SOCS family selectivity of MN551. a** Multiple sequence alignment for SOCS SH2 domains were performed using Clustal Omega and Jalview[69, 70]. Secondary structure elements for SOCS2 are shown above SOCS2 sequence. Cysteine 111 from SOCS2 is highlighted in a purple box and the cysteines from other SOCS proteins are highlighted in green box. **b** MS analysis of SOCS4-EloBC (SBC4) pre-incubated with **MN551** demonstrates that **MN551** does not covalently modify SOCS4. **c** MS analysis of SOCS6-EloBC (SBC6) pre-incubated with **MN551** shows that **MN551** modifies SOCS6 to form a single (18.3%) and minor amount of double (2.6%)

adduct. **d** MS analysis of CISH-EloBC+**MN551**−**MN551** modifies all the CISH to form a single adduct. **e**–**g** Structural superposition of SH2 domains from SOCS2, SOCS4 (PDB: 2IZV), SOCS6 (PDB: 2VIF) and CISH AlphaFold model demonstrate the localization of the EF loops. Unlike SOCS4, SOCS6 does not have a cysteine in its EF loop but has a cysteine 471 in the BG loop that could be engaged by **MN551**. The EF loop in CISH has a cysteine (Cys144) very close to the chloroacetamide group of **MN551** explaining the complete modification of CISH in MS analysis.

Cellular target engagement assay (Cellular Thermal Shift Assay - CETSA) involving transient expression of HiBiT tagged SOCS2 to demonstrate the cellular engagement for our binders[39]. One of the advantages of CETSA is that target engagement can be monitored without the need for the development of a cell-permeable tracer molecule.

For our initial assays, we utilized the permeabilised cell format CETSA to overcome any potential permeability barrier and demonstrate a dose-dependent shift in temperature of aggregation ($T_{agg}$) mediated by **MN551** (Fig. 4a, b). To test the cellular activity of **MN551**, we treated HeLa cells with 50 µM **MN551** for 20 h and performed CETSA in a live-cell mode. We observed a small $T_{agg}$ shift of 2 °C in live mode, which increased to 3−4 °C $T_{agg}$ shift in the permeabilised format (Fig. 4d). We postulate that this observation could likely be a result of the low cellular permeability of **MN551** owing to its negatively charged phosphate group.

To enhance cell membrane penetration, we employed a prodrug approach to mask the phosphate group of **MN551**[6]. At physiological pH

a phosphate group of pY residue is doubly negatively charged and to enhance cell permeability, the negative charge can be masked by enzyme-cleavable hydrophobic groups (Fig. 4c)[6,40]. We decided to make pivaloyloxymethyl (POM) and aryloxy phosphoramidate prodrugs of **MN551** as these prodrug technologies have been proven successful for delivering phosphate-containing molecules inside cells, including phosphotyrosine derivatives[6,41]. Aryloxy phosphoramidate prodrugs of **MN551**, compounds **21-23**, were obtained by treatment of appropriate tyrosine derivative **17** with (chloro(naphthalen-1-yloxy) phosphoryl)-*L*-alaninates as described by Miccoli et al., followed by Boc-deprotection and acylation with chloroacetyl chloride (Supplementary Fig. 4)[41]. POM modified phosphonate **25** (hence afterward referred to as **MN714**) was prepared from tyrosine **17** with the use of bis(POM) phosphoryl chloride (Supplementary Fig. 4)[42].

We utilized CETSA in a live-cell mode to evaluate the efficacy and rank the different prodrugs. We treated live HeLa cells with 50 µM compound (prodrugs) for 20 h and monitored $T_{agg}$ shift as a proxy for the combo of prodrug permeability and unmasking (Fig. 4d). Among

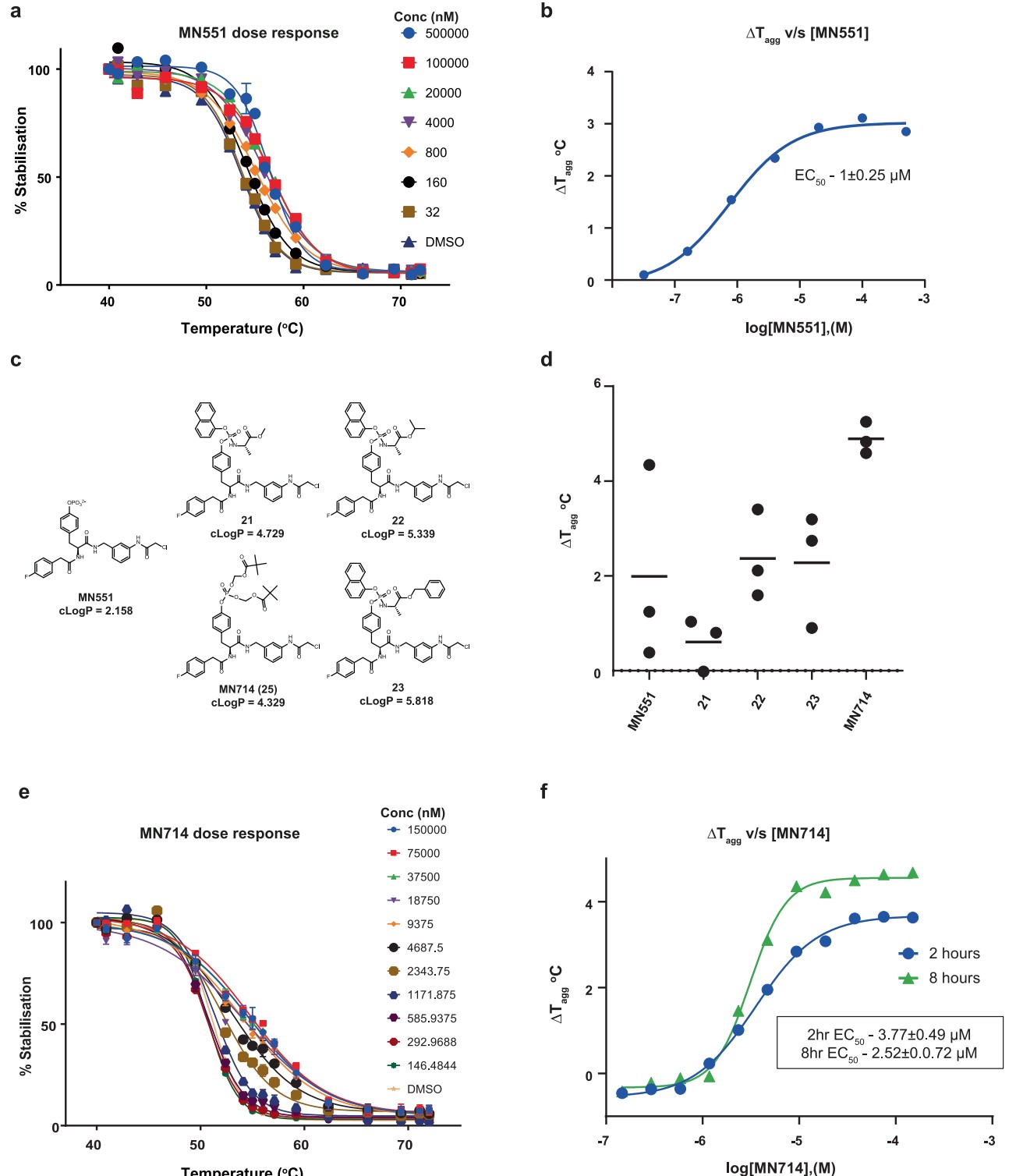

**Fig. 4 | Cellular target engagement assay with split-NanoLuc CETSA.**
**a** Representative curve for **MN551** mediated thermal stabilization of transfected HiBiT-SOCS2 fusion as observed with CETSA in a permeabilised format ($n = 2$). Data are presented as means $\pm$ SD, from technical triplicates. **b** A representative plot of shift in aggregation temperature ($T_{agg}$) against concentration of **MN551**(EC$_{50}$ reported as mean $\pm$ SEM, biological replicates $n = 2$). **c** Chemical structures of prodrugs employed to mask negative charge of **MN551** phosphate group. **d** Live-cell CETSA to assess activity of **MN551** prodrug ($n = 3$). **e** Representative thermal profile of SOCS2 demonstrates efficacy of **MN714** ($n = 3$). Data are presented as means $\pm$ SD, from technical triplicates. **f** Time-dependent reduction in EC$_{50}$ values suggest a covalent nature of SOCS2 engagement (EC$_{50}$ reported as mean $\pm$ SEM, $n = 3$).

all the prodrugs, POM-protected **MN551** (**MN714**), showed consistent superior SOCS2 engagement. The observation could be because the POM group unmasking requires only esterase unlike the reliance of compounds **21–23** on esterase and phosphoramidase for unmasking an aryloxy phosphoramidate group. Furthermore, the increased hydrophilicity of POM-protected **MN714** with respect to the aryloxy phosphoramidate prodrugs likely enhanced the cell permeability. **MN714** (cLogP 4.329) was significantly less lipophilic than all

phosphoramidate prodrugs tested (cLogP ranging from 4.729 to 5.818), placing it firmly within the desired range for lipophilicity as outlined in Lipinski's rule of 5. Having established POM-protected **MN714** as the best prodrug, we demonstrated the dose-dependent cellular potency of the compound. $EC_{50}$ after 2 h and 8 h of treatment is 3.8 μM and 2.5 μM respectively (Fig. 4e, f). The compound, as expected for a covalent binder, displays a time-dependent drop in $EC_{50}$ values.

Although, with the live-cell CETSA setup we could rank the prodrugs for overall potency, we cannot decouple the individual contributions of cellular permeability and prodrug unmasking rate. To understand the rate of unmasking inside the live cells, we developed an in-cell time-resolved ¹⁹F-NMR to monitor the rate of POM group unmasking (Fig. 5). ¹⁹F-NMR is uniquely suited for this task due to the high receptivity of ¹⁹F, low background in biological samples, and remarkable sensitivity of ¹⁹F chemical shifts to their chemical environment[43,44]. The latter means that even the distant fluorine of **MN551** could act as a reporter for the presence of the masking group on the phosphotyrosine moiety. The ¹⁹F 1D NMR spectrum of a K562 cell suspension immediately after treatment with **MN714** shows two broad, partially overlapping peaks 0.110 ppm apart (Fig. 5a). While the downfield peak starts off with higher intensity, it loses intensity quickly and within an hour is overtaken by the growing upfield peak. Within 3 h the downfield peak completely disappears and only the upfield peak is observed. To confirm that this time-dependent change indeed corresponds to the unmasking of **MN714**, the cells were lysed after disappearance of the downfield peak, and a ¹⁹F 1D spectrum of the lysate was collected. A single, sharp peak was observed that increased in intensity when the lysate was spiked with **MN551** (Fig. 5b). In contrast, spiking with **MN714** gave rise to a new peak found 0.112 ppm downfield of the first peak, closely mirroring the peak differences observed in intact cells (Fig. 5c). This suggests that the initially observed downfield peak is indeed caused by **MN714**, while the growing upfield peak corresponds to unmasked **MN551**. The partial overlap of the peaks and poor signal-to-noise prevent exact quantification of the unmasking kinetics, but the data confirms efficient unmasking of **MN714** and suggests a half-life time of 1–1.5 h.

## SOCS2 inhibitor blocks substrate interaction by covalently modifying Cys111 inside the cell

To further validate the cellular potency of our covalent SOCS2 binder and to establish inhibition of SOCS2 binding to its native substrates, we performed a SOCS2 pulldown from K562 cell lysates with biotinylated GHR_pY595 peptide immobilized onto agarose beads[28]. Dose-dependent reduction in the amounts of SOCS2 pulled down when the cell lysates that are preincubated with increasing concentrations of **MN551** demonstrates competitive binding of **MN551** to the GHR substrate binding pocket of SOCS2 (Fig. 6a, b). Pulldown of cellular SOCS2 was abrogated when K562 cells were pre-treated for 6 h with **MN714** prior to cell lysis, further evidencing the inhibitor cell permeability and specific target engagement (Fig. 6c).

While CETSA and GHR pulldown experiments allow ascertaining the engagement of cellular SOCS2, distinguishing reversible binding from covalent binding remains a challenge. Towards evaluating fraction of SOCS2 covalently modified by **MN714**, we performed a SOCS2 immunoprecipitation (IP) in K562 cells treated with **MN714**, followed with tandem MS on the SOCS2 peptides (no significant cytotoxicity was observed with MN714 in K562 cells up to 3-10 μM, see Supplementary Fig. 11). Analysis of IP blots show that the anti-SOCS2 antibody recognizes both apo SOCS2 and **MN551**(unmasked **MN714**) modified SOCS2 efficiently (Fig. 6d). Tandem MS analysis of the IP samples show that **MN551** covalently modifies cellular SOCS2 specifically at Cys111.

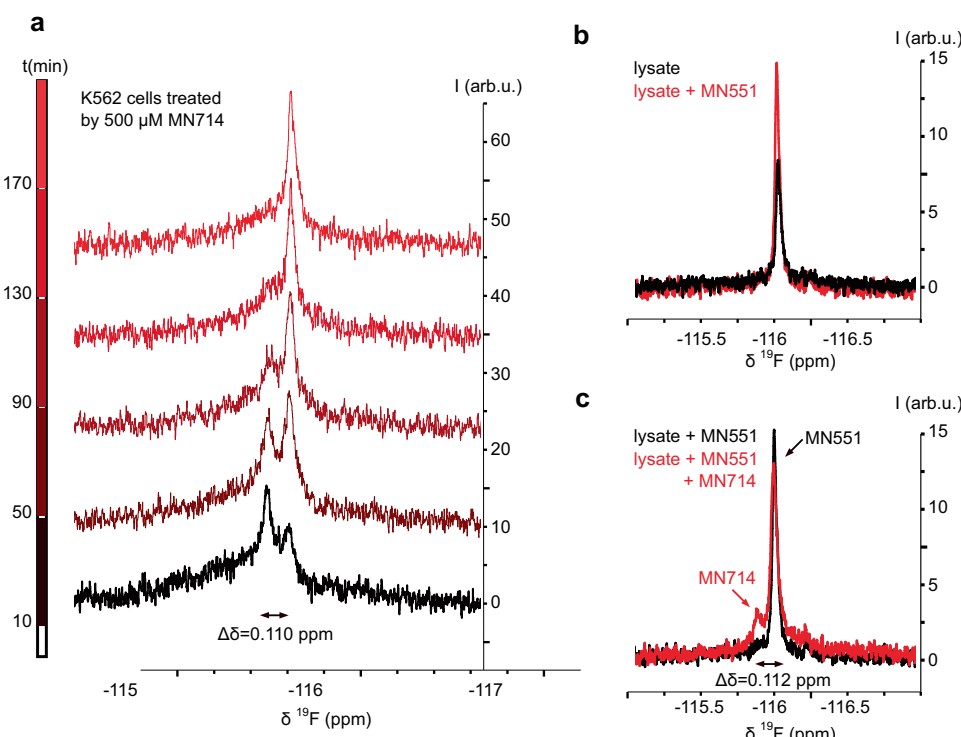

**Fig. 5 | In-cell NMR spectroscopy confirms MN714 prodrug is unmasked inside cells. a** Time-resolved ¹⁹F-NMR of K562 cells treated with **MN714**. Acquisition of the first spectra started 10 min after addition of 500 μM **MN714** and each spectrum was measured over ~40 min. Two broad, overlapping peaks are observed with a difference in chemical shift (δ) of 0.110 ppm. The dominant upfield peak at the beginning of the experiment disappears over time and the downfield peak increases in intensity. **b** ¹⁹F-NMR of the same sample after lysis by freeze-thaw cycles (black). To confirm hydrolysis of the POM group of **MN714** the sample was spiked with 500 μM **MN551** (red). **c** The same sample (**MN551**-spiked, black) was subsequently spiked with **MN714** (red), and a further spectrum acquired. The chemical shift difference between **MN714** and **MN551** in the lysate fits well to the difference between the two peaks observed in cell.

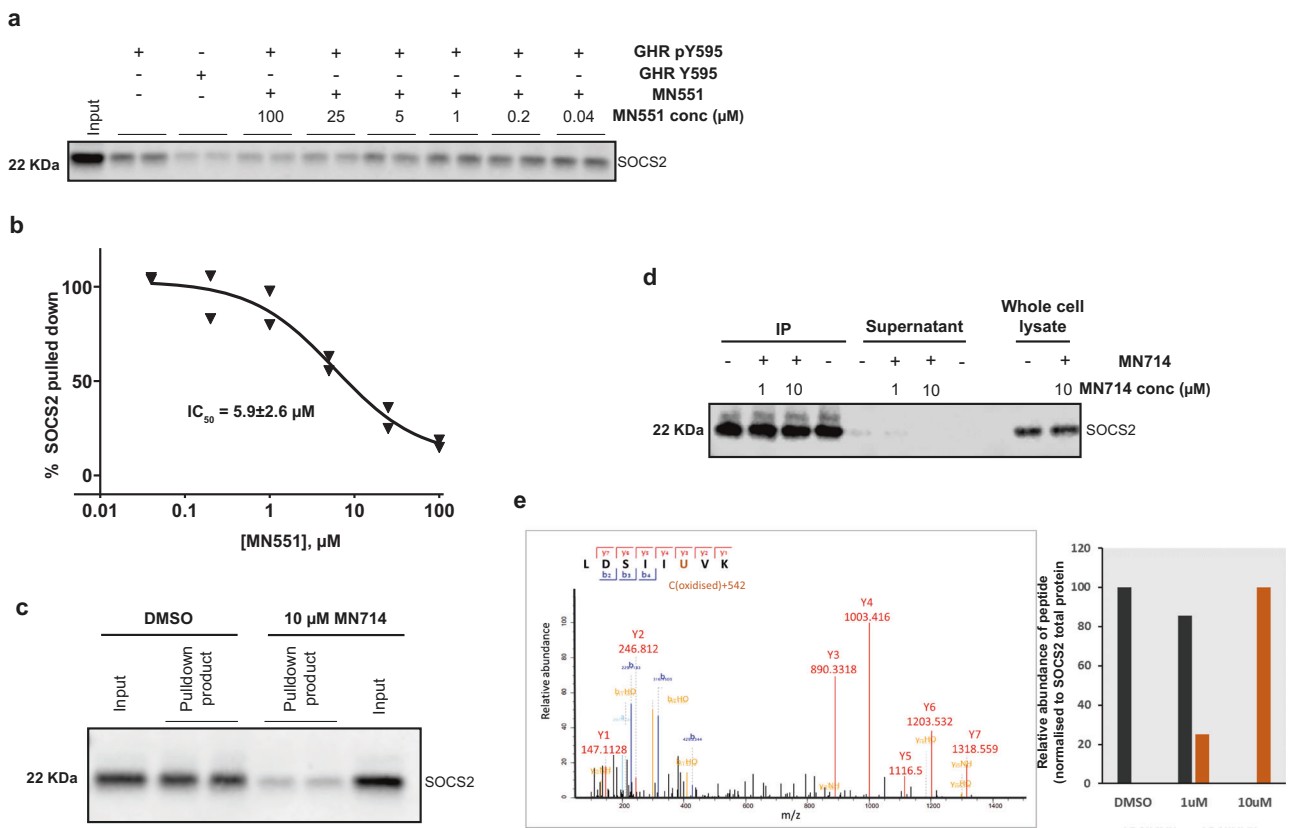

**Fig. 6 | MN551 competitively blocks SOCS2-GHR interaction by covalently engaging Cys111 of SOCS2 inside cells. a** SOCS2 immunoblot of the pulldown product of biotinylated GHR pY595 peptide from K562 cell lysates demonstrates that **MN551** competitively blocks the interaction between SOCS2 and GHR in a dose-dependent manner ($n = 1$). **b** A plot quantifying **MN551** dose-dependent reduction in relative levels of SOCS2 pulled down with biotinylated GHR pY595 (mean ± SEM, technical replicates = 2). **c** SOCS2 immunoblot of the pulldown product with biotinylated GHR pY/GHR peptide from K562 cells pre-treated with 10 μM **MN714** demonstrates effective blockade of SOCS2-GHRpY interaction (representative figure from $n = 2$). **d** Immunoprecipitation blot analysis shows the

presence of SOCS2 protein in SOCS2 IP samples from DMSO control or **MN714** treated K562 cells. Signals of the same molecular weight are also present in the whole cell lysates (WCL without immunoprecipitation) from K562 cells. The supernatant following the respective IP has a very low signal indicating that most SOCS2 is immunoprecipitated. ($n = 1$). **e** Tandem MS analysis of IP sample showed the formation of **MN511** covalently modified peptide. The MS2 spectra confirm the modification on peptide sequence LDSIICVK and mass increase of 542 Da on Cys111. The plot on the right side depicts the relative intensity of modified peptide normalized to SOCS2 total intensity following control or **MN714** treatment and confirming intracellular covalent modification of SOCS2.

Moreover, the MS2 spectra for cells treated with 1 μM and 10 μM **MN714** show a dose-dependent increase in levels of SOCS2 modification (Fig. 6e). An analysis of SOCS2 peptides from cells treated with 10 μM **MN714** shows a complete saturation of Cys111 with no detectable levels of unmodified Cys111.

## Discussion

Drug design efforts targeting SH2 domain have historically been unsuccessful because of enzymatic lability and poor cellular permeability of the negatively charged phosphate group, and the challenges of developing phospho-mimetic ligands. Here, we describe a rational, structure-guided ligand design approach to arrive at a high-affinity cell-permeable ligand for the SH2 domain of SOCS2. There are some important points when considering tackling the main challenges associated with targeting SH2 domains and studying SH2 domain-containing proteins. The phosphate group present in the pY-substrates is typically considered a major liability by medicinal chemists, mainly due to its double-negative charge that hampers cell permeability, and it also presents synthetic challenges[5]. Despite this, there is precedent for phosphate-containing drugs and several therapeutic agents contain phosphates or other phosphorus-containing groups, such as fostemsavir and fosamprenavir for use in patients with HIV infections, fludarabine phosphate for the treatment of chronic lymphocytic leukemia, and fosphenytoin in the treatment of epileptic seizures,

amongst others[45]. Most such phosphate-containing molecules are typically designed as prodrugs to improve the solubility of the parent drug, and undergo hydrolysis during the absorption and distribution process, and prior to entering the cell and reaching the target. In contrast, no pY-containing chemical probes or drugs that target specifically intracellular targets have been developed to our knowledge. In the case of pY-containing molecules, triester prodrugs strategies have been described, but have remained niche[41]. To address the drawbacks of the poor cellular uptake of pY-containing ligands, as prodrug strategy, we evaluated phosphotyrosine-masking groups. Triage of different phosphate-protecting groups allowed the identification of POM as the best prodrug enabling the inhibitor to enter the cell, efficiently unmask, and engage cellular SOCS2 covalently specifically via Cys111 modification. The low efficacy of aryloxy triester phosphoramidate prodrugs can be explained by the lower levels of expression of the enzymes that unmask these prodrugs in HeLa cells[46]. The design and validation methods described in this work exemplify a blueprint that could be in the future used to develop and evaluate other pY-containing small molecules.

The serendipitous observation of Cys111 modification in one of the co-crystal structures solved during the course of the ligand optimization campaign was leveraged to rationally design a covalent inhibitor. Interestingly the same cysteine was identified to be reactive to several electrophiles used by Vinogradova et al., suggesting

tractability of targeting this cysteine covalently[47]. Over the last decade the view on covalent inhibitors has shifted and targeted covalent inhibition has undergone a renaissance as a promising approach in chemical biology and drug discovery, especially for targeting challenging proteins. Covalent modification of the reactive Cys111 residue by **MN551** was confirmed by co-crystal structures, and intact ESI-MS spectra with recombinant protein and mutants. FP-displacement binding assay allowed determining the key kinetic parameter of covalent inhibition−namely $k_{inact}/K_I$. This work provides a benchmark for future optimization strategies to improve the covalency of **MN551**, which will revolve around increasing $k_{inact}/K_I$ by improving the reversible binding affinity and/or increasing the rate of reactivity of the ligand. The chemical probes could provide long-awaited insight into the role of SOCS2 in regulation of growth hormone signaling and NF-kB signaling[48,49]. Future work will also focus on establishing the functional impact of cellular inhibition of SOCS2, and the intra-SOCS family selectivity inhibition, thereby demonstrating full application of **MN551/MN714** and/or their non-covalent analogs as probes to study the JAK/STAT signaling pathway. The efficient covalent modification of recombinant CISH (66-258, Δ174-202) by **MN551** justifies future biophysical, structural, and cellular characterization to verify if the covalency translates to full-length intracellular CISH. Considering the interest in inhibition of CISH as an antitumor therapeutic strategy aimed at enhancing activity of Natural Killer (NK) cells and decreasing PD-1 expression levels, chemical probes targeting selectively CISH would be of great utility[50,51].

Looking forward, the prospect of using a cell permeable, covalent E3 recruiter looks particularly attractive for PROTAC degrader design. SOCS2 is a substrate receptor for a Cullin RING ligase complex, and specifically a Cullin 5 complex (CRL5$^{SOCS2}$), hence should be amenable to be hijacked by targeted protein degraders such as PROTACs. Covalent engagement of the E3 ligase can bring advantages to the sub-stoichiometric catalytic mode of action of the degrader molecule, including simplifying the PROTAC binding equilibrium such that only a single step may occur as the ternary complex dissociates to E3-PROTAC and target protein, but not at the other end as is the case with non-covalent PROTACs[52,53]. These potential benefits of covalency at the E3 ligase can only be realized in full if the half-life of the E3 protein is long enough, such that the covalent modification can be maintained and sustained with time without the engaged compound being rapidly consumed inside the cell. Using cycloheximide treatment to block protein re-synthesis, we showed that SOCS2 has a half-life longer than 24 h (Supplementary Fig. 12), thereby reassuring that **MN714** could offer a kinetic advantage when used as E3 recruiter or handle for PROTACs. PROTACs have provided a pharmacological paradigm of targeted protein degradation, with over 25 compounds today in clinical trials, and many more in pre-clinical development, that degrade a variety of different targets against different diseases[23]. Resistance mechanisms already observed for VHL and CRBN, the most widely hijacked E3 ligases with PROTAC degraders, converge in the loss or missense mutations in E3 ligase components, including the substrate adaptors[54]. These observations motivate the search for new E3 ligase recruiters, as expanding the reach of degraders to more E3 ligases could circumvent such innate or acquired resistance. Since SOCS2 is a substrate receptor for CRL5$^{SOCS2}$, it could open up differentiative biology relative to VHL and CRBN, which are substrate adaptors of CRL2 and CRL4 complexes instead, respectively. Further differences include their subcellular localization e.g. SOCS2 is exclusively cytoplasmic while VHL and CRBN are more ubiquitously expressed, including in the nucleus, and the more tissue-restricted expression profiles of SOCS2 compared to VHL and CRBN, as evident from inspection of protein expression data e.g. from ProteomicsDB (https://www.proteomicsdb.org/). In these regards, work is ongoing in our laboratory to establish proof-of-concept for utilizing the SOCS2 binding ligands developed herein into fast and potent PROTAC

degraders and hence explore the utility of SOCS2 as E3 ligase for targeted protein degradation.

## Methods
### Chemistry
All chemicals unless otherwise stated, were commercially available and used without further purification. Full details of synthetic procedures including Supplementary Figs. 1–4 and NMR spectra of final compound MN714 (Supplementary Fig. 5) are provided as Supplementary Methods.

### Biology
**Cloning and protein expression.** Recombinant SOCS2 (amino acids 32–198), ElonginB (amino acids 1–104) and ElonginC (amino acids 17–112) were co-expressed as previously reported[26,29]. pLIC (His$_6$-SOCS2) and pCDF (EloBC) plasmids were co-transformed into *E. coli* BL21(DE3) and protein expression was induced with 0.5 mM isopropyl β-d-1- thiogalactopyranoside at 18 °C for 16 h. After cell lysis, soluble fraction was purified by affinity chromatography using a HisTrap column (GE Healthcare). Following overnight tag cleavage with tobacco etch virus (TEV) protease, a second HisTrap purification was used to separate cleaved un-tagged SBC2 from His-tagged SBC2. SBC2 was further purified with ion exchange (QHP) column followed with size-exclusion chromatography on a Superdex 75 16/600 column (GE Healthcare) in 25 mM HEPES, pH 7.5, 250 mM NaCl, and 0.5 mM TCEP. SOCS2 mutation C111S (5′-TTGGACTCTATCATATCTGTCAAAT CCAAGCTT-3′; 5′-AAGCTTGGATTTGACAGATATGATAGAGTCCAA-3′; Merck) was introduced using PCR-based site-directed mutagenesis (KOD Hot start DNA Polymerase). SBC2 containing mutant SOCS2 and AviTagged-SOCS2 (5′-CATCATTCTTCTGGTGGCCTGAACGACATCTT CGAGGCTCAGAAAATCG-3′; 5′-CGCCAGACGCGCCGCCTGCGCGCCTT CGTGCCATTCGATTTTCTG-3′, Merck) were purified in the same way as SBC2.

SOCS4 (amino acids 274-437) in a pNIC28-Bsa4 was co-expressed with ElonginB (amino acids 1–104) and ElonginC (amino acids 17–112) at 20 °C with 0.5 mM IPTG for 18 h and the resulting ternary complex SOCS4/EloBC (SBC4) purified the same way as SBC2. The TEV-digest and second affinity column were omitted as the tag proved to be non-cleavable.

SOCS6 (amino acids 353-535) in a pGTVL2 vector with a TEV-cleavable N-terminal His6-GST-tag was co-expressed with ElonginB (amino acids 1–104) and ElonginC (amino acids 17–112) as above. Expression was induced at 16 °C for 22 h and the resulting ternary complex SOCS6/EloBC (SBC6) purified by affinity chromatography using a HisTrap column (GE Healthcare). The tag was removed by TEV treatment and a second HisTrap purification, followed by size-exclusion chromatography on a Superdex 75 16/600 column (GE Healthcare) in 25 mM Tris pH 7.5, 250 mM NaCl, and 0.5 mM TCEP.

CISH (66-258 Δ174-202) was codon-optimized for bacterial expression and cloned into a pGEX-6P-1 vector[38]. It was co-expressed with ElonginB (amino acids 1–104) and ElonginC (amino acids 17–112) at 18°C and 1 mM IPTG in BL21(DE3). The CISH/EloBC complex was purified using gluthathione beads in 50 mM Tris, pH 7.5, 300 mM NaCl, 20 mM sodium sulfate, 1 mM TCEP, 10 % glycerol, eluted by on-column cleavage with prescission protease and further purified by SEC on a Superdex S75 16/600 in 50 mM Tris, pH 7.5, 300 mM NaCl, 20 mM sodium chloride, 1 mM TCEP, 5 % glycerol.

**Crystallization of SOCS2-EloBC (SBC2): soaking and co-crystallization.** Prior to crystallization, the SBC2 protein was buffer exchanged into 137 mM NaCl, 2.7 mM KCl, 10 mM Na$_2$HPO$_4$, and 1.8 mM KH$_2$PO$_4$ (1× PBS) buffer with 0.5 mM TCEP. For soaking experiments SBC2 apo crystals were obtained by crystallizing 17 mg/ml SBC2 in 100 mM tris(hydroxymethyl)aminomethane (TRIS)- *N, N*-Bis(2-hydroxyethyl) glycine (BICINE) pH 6.5–7, 9–14% polyethylene glycol (PEG) 8 K,

26–20% ethylene glycol at 4 °C using the sitting-drop method (0.4 μl of protein + 0.35 μl reservoir solutions + 0.05 μl of seed stock of disrupted crystals). Ligands were soaked overnight into crystals that were obtained after ~5–14 days. Co-crystal complexes of SBC2-ligand were obtained by incubating 17 mg/ml SBC2 with 0.5 mM compound **13/MN551** at 4 °C overnight prior to crystallization in 17.5% PEG 3350, 0.1 M BTP pH 6.5, 0.178 M sodium citrate tribasic dihydrate at 4 °C using the sitting-drop method (0.4 μl of protein + 0.4 μl reservoir solutions). Soaked SBC2 crystals were flash-frozen without any additional cryoprotectant whereas the co-crystals were frozen in presence of 20% ethylene glycol as cryoprotectant. Diffraction data were collected at Diamond Light Source beamline i03/i04 or The European Synchrotron Radiation Facility ID30-A1 and images were processed with Xia2 Dials/autoPROC[55–59]. The structure was solved by molecular replacement using SBC2 crystal structure (PDB entry 2C9W) as a search model. Subsequent iterative model building, and refinement was done according to standard protocols using COOT and Phenix. Ligand restraints were generated using the Phenix eLBOW/PRODRG server[60–63]. Data collection and refinement statistics for all structures are described in Supplementary Table 1.

**$^{19}$F CPMG NMR spectroscopy.** Experiments were performed using an Avance III 500 MHz Bruker spectrometer equipped with a 5 mm CPQCI 1H/19 F/13 C/15 N/D Z-GRD cryoprobe at 298 K as described previously[29]. Spectra were recorded using 80 scans of a CPMG pulse sequence that attenuates broad resonances. A CPMG delay of 0.133 s was used, to maximize the difference between the signal intensity of spy molecule alone and in the presence of protein. The transmitter frequency was placed close to the resonance of $O_1 = −35451$ Hz (−75.3 ppm). Protein was used at 5 μM, spy molecule was used at 100 μM, compounds were tested at 100 μM in buffer containing 20 mM HEPES pH 8, 50 mM NaCl, 1 mM TCEP, 20% $D_2O$, 2% DMSO. All NMR data were processed and analyzed using TopSpin (Bruker). The dissociation constant was calculated as described previously[29].

**Surface Plasmon Resonance (SPR) binding studies.** SPR experiments were performed using a Biacore S200 instrument (GE Healthcare) in 20 mM HEPES pH 7.5, 150 mM NaCl, 1 mM TCEP, 0.005% Tween20, 2% DMSO buffer at 10 °C. Biotinylated SBC2 was immobilized onto a chip surface (the final surface density of biotinylated SBC2 was ~3000–4000 RU). AviTagged-SBC was biotinylated using BirA enzyme as per manufacturer's protocol (Avidity). Compounds were serially diluted in the running buffer and injected individually: contact time 60 s, flow rate 30 μL/min, dissociation time 150 s, using a stabilization period of 30 s and syringe wash (50% DMSO) between injections. Data analysis was carried out using Biacore Evaluation Software (GE Healthcare). All data were double-referenced for reference surface and blank injection. The processed sensograms were fit to a steady-state affinity using a 1:1 binding model for $K_D$ estimation.

**ESI-MS of recombinant protein.** SBC2 protein (40 μM), SBC2 C111S, SBC4, SBC6, or CISH were incubated with an equimolar concentration of **MN551** in 50 mM Tris, pH 8.0, 50 mM NaCl, 1 mM TCEP. At defined time points 20 μl samples were removed and the protein precipitated by the addition of 80 μl methanol. The precipitated protein was pelleted by centrifugation, washed with 500 μl methanol, and resuspended in an aqueous solution of 15% acetonitrile and 1% TFA. Samples were separated by HPLC on a C3 column using a 10–75% gradient of acetonitrile and analyzed using an Agilent 6130 quadrupole MS. Spectra were deconvoluted and integrated using Agilent LC/MSD ChemStation.

**Differential Scanning Fluorimetry (DSF).** DSF experiments were performed on a Biorad CFX96 RT-PCR machine. 5 μl of SBC2 were incubated with 100 μl **MN551** (2 % DMSO) or DMSO control in 50 mM Tris, pH 8.0, 50 mM NaCl and 6.6 × SYPRO Orange. The temperature was ramped up in 1 °C steps between 25 and 95 °C with 30 s incubation at each step. Melting curves were analyzed by determining the minimum of the first derivative using the Biorad CFX Manager software.

**Isothermal titration calorimetry (ITC).** Experiments were performed with ITC200 instrument (Malvern) in 100 mM HEPES pH 7.5, 50 mM NaCl, 0.5 mM TCEP at 298 K. The ITC titration consisted of 0.4 μl initial injection (discarded during data analysis) followed by 19 of 2 μl injections at 120 s interval between injections. 1 mM of the ligands in 2–4% DMSO were titrated into 100 μM SBC2. The GHR_pY595 peptide sequence was used for the ITC run−PVPDpYTSIHIV-amide[29]. For the competition run, 1 mM GHR_pY595 was titrated into 100 μM SBC2 that was preincubated with 200 μM **MN551** at room temperature for 2 h. Binding data was subtracted from a control titration where ligand was titrated into buffer, and fitted using a one-set-of-site binding model to obtain dissociation constant ($K_d$), binding enthalpy (ΔH), and stoichiometry (N) using MicroCal PEAQ-ITC Analysis Software1.1.0.1262 and MicroCal ITC-ORIGIN Analysis Software 7.0 (Malvern).

**Fluorescence Polarization assay.** To monitor the SBC2-ligand binding kinetics we designed a fluorescein-labeled small-molecule probe - compound **26**. The fluorescence polarization (anisotropy) of a three-fold serial dilution of 50 μM SBC2 in 10 nM of the probe in 100 mM HEPES pH 7.5, 50 mM NaCl, 0.5 mM TCEP, 2% DMSO was measured to determine the $K_L$ (Supplementary Fig. 9A) (BMG Labtech PHERAstar – firmware v1.33). Protein titration and the curves were fitted using GraphPad Prism 9, using one-site fitting models to determine the probe binding constant, $K_L$. Samples were run in triplicate in 384 well plates, using a total volume per well of 16 μL. The bound fraction of the probe is measure using the equation:

$$F_b = \frac{A_f - A}{A_f - A_b} \qquad (1)$$

Where,

$A$−measured anisotropy.

$A_f$−free anisotropic value for the fluorescent probe

$A_b$−bound anisotropic value for the fluorescent probe

A time-dependent reduction in probe anisotropy were measured for a 2-fold serial dilution of 100 mM compound **MN551** in 10 nM probe +300 μM SBC2. To determine the initial reversible binding affinity, the initial bound fraction of the probe at time = 0, $F_{bo}$ (the y-intercept in $F_b$ v/s Time plot, Supplementary Fig. 9B) was plotted against **MN551** concentration, and the obtained $IC_{50}$ is then converted to $K_i$ (Supplementary Fig. 9C)[17]. The slope from $F_b$ versus time graph when divided by $F_{bo}$ gives rate of covalency, $k_{obs}$. A plot of $k_{obs}$ v/s **MN551** concentration using Michaelis−Menten model of GraphPad Prism 9, provides the parameters $k_{inact}$ (maximum potential rate of inactivation) and $K_I$ (concentration of **MN551** at which $k_{obs} = k_{inact}/2$) (Supplementary Fig. 9D, E).

The observed first-order rate constant, $k_{obs}$ is related to $K_I$ and $k_{inact}$ by the equation:

$$k_{obs} = \frac{k_{inact}[I]}{K_I + [I]} \qquad (2)$$

Since $k_{inact}/K_I$ determines the overall covalent efficiency, to obtain a precise estimation of this parameter, we repeated the anisotropy measurements with a 1.4-fold serial dilution of 1 μM compound **MN551** in 10 nM probe + 300 μM SBC2. At [I]«[$K_I$] the above equation is converted to

$$k_{obs} = \frac{k_{inact}}{K_I} \qquad (3)$$

The slope from the plot of $k_{obs}$ against concentration of **MN551** gives us the covalent efficiency parameter $k_{inact}/K_I$ (Supp Fig. 9F)[30,33].

**Structural modeling of CISH.** The structure of CISH in complex with ElonginB and C was predicted using Alphafold-Multimer[64]. The sequence of either the expression construct (CISH 66–258 Δ174–202, ElonginB 1–104, and ElonginC 17–112) or full-length CISH was used. The full database of reference sequences and structures was used with structures published after 01-01-2022 excluded and five seeds per model were generated. The best prediction by pLDDT was used for a final relaxation step to improve local geometry.

**Cell lines and culture.** Cell lines were obtained through American Type Culture Collection (ATCC) and tested weekly for mycoplasma contamination using MycoAlert Mycoplasma detection kit (Lonza). HeLa and K562 cells were grown in Dulbecco's Modified Eagle's Medium (DMEM) medium (Gibco) and Iscove Modified Dulbecco Media (IMDM) (Gibco) media respectively. The media was supplemented with 10% FBS (Gibco), L-glutamate (Gibco), penicillin, and streptomycin.

**Split-NanoLuc cellular target engagement assay.** N-terminal HiBiT-SOCS2 fusion was generated by encoding SOCS2 (32–198) in pBiT3.1-N expression vector (Promega). The transfection and target engagement assay were performed by modifying a NanoLuc target engagement protocol (Promega) with some small modifications[65]. Briefly, HiBiT-SOCS2 fusion construct was transfected into HeLa cells using FuGENE HD (Promega) according to the manufacturer's protocol. HiBiT-SOCS2 fusion construct was diluted into Transfection Carrier DNA (Promega) at a mass ratio of 1:100 or 1:10 for permeabilised and live-cell CETSA respectively. FuGENE HD was added at a ratio of 1:3 (µg DNA: µL FuGENE HD). 1 part (vol) of FuGENE HD complexes thus formed were combined with 20 parts (vol) of HeLa cells suspended at a density of $2 \times 10^5$ cells/mL, followed by incubation in a humidified, 37 °C + 5% $CO_2$ incubator. After 20 h of incubation HeLa cells were detached and resuspended at a density of $2.5 \times 10^5$ cells/mL in Opti-MEM media without serum or phenol red but supplemented with protease inhibitor cocktail (Roche). Compounds stocks were added to cell suspension to maintain a final DMSO concentration of 1%. For the permeabilised assay format, 96-well PCR plates (Thermo) were pre-treated with digitonin to have a final well concentration of 50 µg/ml. 50 µl Opti-MEM cell suspension was then aliquoted to each well of the 96-well plate. The plates are then covered with microporous tape sheet (Qiagen) prior to incubation in a humidified, 37 °C + 5% $CO_2$ incubator. Post-incubation we heat the PCR plates in 16-point temperature (split across two runs) curve spanning degrees 40-72 °C temperature span (CFX96 Touch Real-Time PCR Detection System, BioRad). The thermal cycler block was allowed to rise to the designated temperatures prior to placement of the PCR plates into the heating block and heat treatment for 3 min. After incubating the plates at room temperature for over 3 min, 50 µl of the Nano-Glo HiBiT Lytic mix (Promega) was added to each well. Samples were mixed by pipetting and placing the plate on an orbital shaker (600 rpm) for 5 min. 20 µl of the samples (in triplicates) were transferred to a AlphaPlate-384 plate (Perkin Elmer). Luminescence was recorded with GloMax Discover Microplate Reader (Promega). To generate apparent $T_{agg}$ curves, the data is first converted to percent stabilized by relating the observed luminescence to the luminescence of the lowest temperature (40 °C) for that given sample. Data are then fitted to obtain apparent $T_{agg}$ values using the Boltzmann Sigmoid equation using Graphpad Prism 9.

**In-cell NMR spectroscopy.** K562 cells were maintained in IMDM media supplemented with 10% FBS. For NMR measurements cells were harvested by centrifugation, washed, and resuspended in Opti-MEM serum-free media to obtain a suspension with 65% v/v cell content. 10% $D_2O$ and 500 µM prodrug in DMSO-d6 were added (0.5% final DMSO

concentration) and incubated on a spinning wheel for 5 min. A 200 µl suspension was transferred into a 3 mm Shigemi tube (without insert) matched to the magnetic susceptibility of water. 1D $^{19}F$ spectra with inverse-gated decoupling of protons were collected at 37 °C with 2048 scans (experiment time 37 min) on a Bruker AVANCE III spectrometer with an 11.7 T magnet using a QCI-F cryoprobe at 470 MHz Lamour frequency. After 4 h the sample was lysed by freeze-thawing and a further spectrum of the lysate was collected. To confirm the identity of the observed species, the lysate was sequentially spiked with 500 µM **MN551** and prodrug and measured again. For comparison, 1D $^{19}F$ spectra of pure **MN551** and prodrugs were collected in Opti-MEM media with 10% $D_2O$ and 0.5% DMSO-d6. All spectra were processed and analyzed using Topspin 4.1.1.

**SOCS2 pulldown with biotinylated-GHR_pY595 peptide.** K562 cells were treated with either DMSO or 10 µM **MN714** for 6 h before being lysed. For experiments with **MN551**, DMSO-treated lysate was used. DMSO-treated K562 lysate (1 mg/ml) was pre-cleared using control agarose beads (Pierce Control Agarose Resin, 26150) which were washed thrice using 50 mM Tris pH 7.4, 150 mM NaCl buffer. **MN551** (100 µM, 25 µM, 5 µM, 1 µM, 0.2 µM and 0.04) was added to 75 µg/100 µl pre-cleared lysate containing protease (Merck, 11873580001) and phosphatase inhibitors (Sigma, 524627 in Tris-NaCl buffer). **MN551** was incubated with the pre-cleared lysate for 1.5 h at room temperature. Pulldown experiments were performed using biotinylated peptides, GHR pY595 (Biotin-aminohexanoic acid-PVPDpYTSIHIV-amide) or GHR Y595(Biotin-aminohexanoic acid-PVPDYTSIHIV-amide). The peptides were immobilized using 30 µl/IP high-capacity streptavidin agarose beads (Pierce High-Capacity Streptavidin Agarose, 20357) by incubating for 30 min a 4 °C. Separate tubes of peptide incubation were performed for GHR pY595 and GHR Y595. Unbound peptide was washed off through 3× cycles of buffer wash and spin. Beads were resuspended to a total volume of 100 µl per IP sample. 100 µl beads with immobilized peptides was added to the **MN551** lysate tubes. Samples were incubated for 40 min at room temperature. Samples were spun and unbound supernatant was removed. Beads were washed 3× with 0.05% NP40, 50 mM Tris pH 7.4, 150 mM NaCl buffer before being resuspended in 35 µl 1× LDS. Samples were heated for 10 min at 95 °C after which the elute was transferred to a fresh tube. 20 µl of each sample was analyzed by Western Blot (4-12% Bis-Tris gel 180 V, 60 mins) before blotting for SOCS2 (ab109245, 1:1000 dilution) overnight at 4 °C. Uncropped blot images collected with ChemiDoc Touch imaging system (BioRad) operated by Image Lab (v2.4.0.03) and antibody validation provided in Source Data. The following secondary antibodies used were IRDye® 800CW anti-rabbit (no. 926-32211, LiCor, 1:10000 dilution), and hFABTM rhodamine anti-tubulin (no. 12004165, Biorad, 1:5000 dilution). Western blot images processed with Image Lab (BioRad).

**Immunoprecipitation and sample processing for LC-MS/MS.** Cells were treated with either DMSO, 1 µM, or 10 µM **MN714** for 6 h in 10 cm dish. Cells were harvested, washed with PBS, and lysed in 500 µl ml RIPA buffer (Sigma-R0278: 150 mM NaCl, 1.0% IGEPAL® CA-630, 0.5% sodium deoxycholate, 0.1% SDS, 50 mM Tris, pH 8.0, and freshly added protease inhibitor tablet). Cellular debris was pelleted, the lysate supernatant was aspirated to a new tube and subjected to immunoprecipitation at 4 °C for 4 h with 3 µl anti-SOCS2 antibody coupled to 20 µl of protein A/G beads (abcam- ab109245). The beads were washed 3 times with 1× RIPA buffer and precipitated protein samples were eluted with 1 × NuPAGE™ LDS sample buffer, and 20% of the sample was analyzed by SDS-PAGE and western blot using a rabbit anti-SOCS2 primary antibody (abcam) and IRDye® 800CW donkey anti-rabbit IgG secondary antibody (abcam). The remaining precipitated protein samples were processed for mass-spec analysis. In brief, the samples were reduced with DTT alkylated with Iodoacetamide and subjected to the SP3

protein clean-up procedure[66]. The samples were eluted from SP3 beads into digestion buffer (0.1% SDS, 50 mM TEAB pH 8.5, 1 mM CaCl$_2$) and digested with trypsin at a 1:50 enzyme to protein ratio, and peptide clean up were performed according to the SP3 protocol. Samples were eluted into 2% DMSO and dried under vacuum. The dried samples were suspended in 40 μl 1% formic acid (FA) and submitted for LC-MS/MS run.

**Sample analysis by LC-MS/MS.** LC-MS analysis was performed by the FingerPrints Proteomics Facility (University of Dundee). Analysis of peptide readout was performed on a Q-Exactive™ plus, Mass Spectrometer (Thermo Scientific) coupled to a Dionex Ultimate 3000 RS (Thermo Scientific). LC buffers used are the following: buffer A (0.1% formic acid in Milli-Q water (v/v)) and buffer B (80% acetonitrile and 0.1% formic acid in Milli-Q water (v/v)). 15 μl of each sample were loaded at 10 μL/min onto a trap column (100 μm × 2 cm, PepMap nanoViper C18 column, 5 μm, 100 Å, Thermo Scientific) equilibrated in 0.1% TFA. The trap column was washed for 3 min at the same flow rate with 0.1% TFA and then switched in-line with a Thermo Scientific, resolving C18 column (75 μm × 50 cm, PepMap RSLC C18 column, 2 μm, 100 Å). The peptides were eluted from the column at a constant flow rate of 300 nl/min with a linear gradient from 2% buffer to 5% in 5 min, from 5% B to 35% buffer B in 125 min, and then to 98% buffer B within 2 min. The column was then washed with 98% buffer B for 20 min. Two blanks were run between each sample to reduce carry-over. The column was kept at a constant temperature of 50 °C.

The Q-exactive plus was operated in data dependent positive ionization mode. The source voltage was set to 2.30 kV and the capillary temperature was 250 °C.

A scan cycle comprised MS1 scan (m/z range from 350–1600, ion injection time of 20 ms, resolution 70000 and automatic gain control (AGC) 1 × 106) acquired in profile mode, followed by 15 sequential dependent MS2 scans (resolution 17500) of the most intense ions fulfilling predefined selection criteria (AGC 2 × 105, maximum ion injection time 100 ms, isolation window of 1.4 m/z, fixed first mass of 100 m/z, spectrum data type: centroid, intensity threshold 2 × 104, exclusion of unassigned, singly and >7 charged precursors, peptide match preferred, exclude isotopes on, dynamic exclusion time 45 s). The HCD collision energy was set to 27% of the normalized collision energy. Mass accuracy is checked before the start of samples analysis.

**Data Evaluation using MaxQuant.** MS raw data were analyzed using MaxQuant software, standard settings were used with the following changes and additions: The modified FASTA databases with individual substitutions of cysteines with the placeholder "U" (selenocysteine) were used[67]. **MN551** covalently modified cysteine (C(26)H(25)N(3)O(7)FPSe(-1)S) was set on the placeholder amino acid "U". In MaxQuant settings, acetylation (Protein-N-terminus), oxidation of methionine (M), and **MN551** modified Cysteine (on placeholder U) are set as variable modifications. Carbamidomethyl (C$_2$H$_3$NO) was set as a fixed modification on cysteine. The digestion enzyme was set to Trypsin/P with maximum number of missed cleavages of 2. The relative intensity of peptide LDSIICVK without and with a modified cysteine (placeholder "U") were analyzed by normalizing to total intensity of SOCS2 protein.

**Glutathione (GSH) reactivity assay. MN551/MN714** (1/10 μM) were incubated at 37 °C with 5 mM GSH in 100 mM potassium phosphate buffer, pH 7.4 (0.1% DMSO). The analytes were separated with ACQUITY UPLC BEH C18, 1.7 μm 2.1 × 50 mm Column (Part No.186002350), and the mobile phase consisted of 0.1% formic acid in water/acetonitrile with a linear gradient of organic phase. Samples were then analyzed with an LC (Shimadzu LC-40)/MS/MS (Triple Quad 6500+). From the mass spectrum, peak area ratios (peak area analyte/peak area of a stable internal standard) were calculated and the

percent compound remaining was determined relative to time zero. Rate of disappearance of analyte ($k_e$) and half-life ($T_{1/2}$) were calculated by fitting to a pseudo-first-order kinetic equation

$$C_t = C_0 * e^{-k_e * t} \qquad (4)$$

$$C_t = \frac{1}{2} C_0 \qquad (5)$$

$$T_{\frac{1}{2}} = \frac{Ln2}{-k} = \frac{0.693}{-k} \qquad (6)$$

$C_t$–Concentration of analyte after time, $t$
$C_0$–Concentration of analyte at time = 0

### Reporting summary
Further information on research design is available in the Nature Portfolio Reporting Summary linked to this article.

## Data availability
X-ray crystallographic data have been deposited to the PDB under accession codes 7ZLP (compound **9** soaked in complex with SBC2), 7ZLN (compound **11** soaked in complex with SBC2), 7ZLO (compound **12** soaked in complex with SBC2), 7ZLR (compound **13** soaked in complex with SBC2), 7ZLS (compound **13** co-crystallized in complex with SBC2), 7ZLM (compound **MN551** co-crystallized soaked in complex with SBC2). NMR spectra for **MN714** are provided in the Supplementary Information. All other data generated for all Tables, Figures, and Supplementary Figures are available in the Supplementary Data files. Plasmids generated in this study are available from the corresponding authors upon request. Source data are provided with this paper.

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

## Acknowledgements

We thank Wei-Wei Kung for her contributions during the inception of the project, and Jane Wright for validating the SOCS2 antibody; Dalia Barsyte-Lovejoy, Magdalena Szewczyk, and Santha Santhakumar from the University of Toronto for their support and discussions on the split NanoLuc CETSA assay; Matt Clifton and Fred Cohen from Nurix Therapeutics for their valuable feedback and suggestions in the early phases of the project; Dougie Lamond and the Dundee FingerPrints Proteomic facility for support with the proteomic analysis; Edwin Allen and the MRC protein phosphorylation and ubiquitylation unit (MRC-PPU) tissue culture team for technical support with tissue culture; and Satpal Virdee and Mathieu Soetens (MRC-PPU) for providing access and assistance with the Agilent system for intact ESI-MS. Research reported in this publication was supported by the European Research Council (ERC) under the European Union's Seventh Framework Programme (FP7/2007-2013) as a Starting Grant (grant agreement no. ERC-2012-StG-311460 DrugE3CRLs to A.C.), and the Innovative Medicines Initiative 2 (IMI2) Joint Undertaking under grant agreement no. 875510 (EUbOPEN project), and Nurix Therapeutics. Biophysics and drug discovery activities at Dundee were supported by Wellcome Trust strategic awards to Dundee (100476/Z/12/Z and 094090/Z/10/Z, respectively). We thank the Diamond Light Source beamline i03/i04 and the European Synchrotron Radiation Facility ID30-A1 for the beamtime.

## Author contributions

S.R. and N.M. contributed equally. S.R., N.M., and A.C. conceived and planned this project. S.R. and N.M. designed and conducted experiments, compiled and analyzed data, and prepared figures. N.M., A.T. designed and synthesized chemical compounds. E.B. contributed synthesis of compounds, with input from R.G. D.L. and R.C. characterized, optimized, and scaled up the synthesis of compounds and contributed text to the paper. N.M. designed and performed NMR and SPR binding experiments. S.R. designed and carried out protein crystallography, ITC, FP, and CETSA experiments. K.H. designed and performed the in-cell NMR, nanoDSF, ESI intact MS, and alphafold modeling. M.N. designed and performed the SOCS2 IP and LC-MS/MS. M.N. and B.F. performed the competitive peptide-IP experiment. A.C. designed and supervised the project. S.R., N.M., and A.C. co-wrote the manuscript with input from all co-authors.

## Competing interests

The Ciulli laboratory receives or has received sponsored research support from Almirall, Amgen, Amphista Therapeutics, Boehringer Ingelheim, Eisai, Merck KGaA, Nurix Therapeutics, Ono Pharmaceutical and Tocris-Biotechne. A.C. is a scientific founder, shareholder, and advisor of Amphista Therapeutics, a company that is developing targeted protein degradation therapeutic platforms. A.T. and N.M. are currently employees of Amphista Therapeutics. The remaining authors declare no competing interests.
