## [Peer Review File · Nature Communications]

Reviewers' Comments:

Reviewer #1:

Remarks to the Author:

Ramachandran et al. present an exciting manuscript in which they developed a phospho-tyrosine (pY)-masked covalent ligand that targets the SH2 domain of the SOCS2 E3 ligase. SOCS family proteins are central negative regulators of cytokine signaling pathways and some members are emerging targets for cancer immune therapies. SH2 domains are notoriously difficult to target and little progress to develop selective, potent, metabolically stable and cell permeable binders/inhibitors were made over the past two decades despite considerable efforts in big pharma, biotech and academia. In this paper, starting with masked pY ligand fragments and through a med chem tour-de-force, the authors develop SOCS2-targeting compounds with binding affinities in the hundred nM range. Co-crystal structures identify serendipitous covalent engagement of the lead compound (cpd 13) with a Cys in the EF loop of the SOCS2 SH2. Exchange to a more reactive covalent warhead resulted in MN551 that is subsequently characterized with great scientific rigour and depth (and at a high standard that is typically only found in preclinical drug discovery departments of major biotech/pharma companies). Cellular target engagement is demonstrated as well as unmasking of the pY group in cells. Overall, this is an extremely exciting paper that convincingly reports an SH2 domain-targeting compound that covalently binds SOCS2 in cells and has high enough potency to serve as a SOCS2 inhibitor. In my opinion, this is the first report of an SH2 domain inhibitor that combines sufficient potency, cellular activity and high selectivity (through covalent engagement of target-selective cysteine). As discussed by the authors, this work has important implications for expand PROTAC/molecular glue degraders besides utility of the developed compound to study and perturb cytokine signaling pathways.

I already read a previous version of the manuscript in bioRxiv with great interest and excitement and was very happy when I was asked to review the manuscript for Nature Communications.

There are some minor points that I would like the authors to answer:

1. Selectivity within SOCS family: Selectivity for SOCS4 and SOCS6 is studied (Fig 3), but I wonder what the affinity for CISH is? The length of the EF loop in CISH and SOCS2 is identical, the sequence is conserved with two positive aa in the C-terminal part of the loop and the Cys is at the same relative position in the two protein. There is no CISH-EloB/C crystal structure published but the recombinant protein was produced previously. Please comment.
2. Does MN551 act as an pY peptide competitor? Previous work of the Ciulli lab showed high-affinity binding and the structural basis for EpoR and GHR binding of SOCS2. Is MN551 competing with these pY peptides, e.g. in an FP assay in vitro?
3. Although given the large amount of fantastic data, I feel slightly embarrassed to ask this: Do the authors have some preliminary data that they would want to include to show activity of MN551 in a cellular read-out, e.g. hyperactivation of a cytokine signaling dependent readout?

Reviewer #2:

Remarks to the Author:

The manuscript by Ciulli and colleagues describes the discovery of a selective covalent inhibitors of the E3 ubiquitin ligase SOCS2. The work covered the design, synthesis, structural biology, metabolism and cell activity of the compounds. Overall, this is a rather comprehensive manuscript that builds on recent progress on the discovery and cell delivery of pY containing drugs. The use of pY prodrug strategies to improve the cellular uptake of the compounds is notable. As mentioned above this is a rather comprehensive manuscript, but few comments for the authors to consider.

- The authors summarised the functions and signalling of SOCS proteins well. However, many will find this rather detailed and thus the authors should include a Figure showing JAK/STAT signalling and SOCS.
- Again, the text description of the SOCS2 structure binding to the phosphopeptides was very good, but a Figure showing this would make it easier to understand and visualise the key interactions with pY.
- The authors need to include the calculated logP values of the compounds and integrate these in the discussion in terms of cell permeability. It is common that the prodrugs can be associated with

high lipophilicity, which limits their water (cell media) solubility when treating cells.

- Page 7, lines 234-235, sentence "A co-crystal structure of SBC...of MN551 and cyc111" needs to be followed by the Figure number referring to the co-crystal structure and mass spectrometry data.
- In the discussion around the pY prodrugs, the authors need indicate that the aryloxy triester phosphoramidate prodrugs are generally poorly metabolised in HeLa cells due to the low expression of the enzymes that metabolize these prodrugs in HeLa cells. See Miccolli et al <https://doi.org/10.1021/acs.jmedchem.1c01490> who examined the enzymes expression across a selection of cells including HeLa cells.
- Also, in describing the pY prodrugs data, the authors need to comment on the incubation time being 20h, which is suitable for POMtide prodrugs, but the aryloxy triester prodrugs are often incubated for longer times > 24 h to see some effect.
- Although in the discussion the authors focused on the potential future use of MN551/714 in the design of PROTACs, an equal emphasis and future outlook on the use of MN551/714 alone to modify SOCS2 (JAK/STAT signalling) is needed since covalent inhibitors are being widely pursued in drug discovery.
- Finally, the figures resolution is poor; authors needs to have high resolution figures.

Together, this manuscript and after the corrections will be provide a useful blue print for the future discovery of SH2 domain inhibitors.

Reviewer #3:

Remarks to the Author:

The manuscript by Ramachandran and colleagues, describes the chemical development of a small chemical binder of the SOCS2-SH2 domain.

SH2 domains have traditionally been difficult to drug, based on the common mode of binding across the SH2 family, and the requirement for a negatively charged phospho-tyrosine that impedes membrane permeability. In addition, once inside the cell, there is the potential for de-phosphorylation. The authors overcame these constraints by generating a covalent pro-drug based on pTyr.

This study represents a significant advance in the field, given the challenges in targeting SH2 domains, and the limited number of binders described to date that target the SH2-pTyr binding pocket. This is the first description of a binder targeting the pTyr pocket in the SOCS family proteins.

This is a well-written, comprehensive and informative study. However, it is not clear whether the prodrug MN714 will act as an inhibitor of SOCS2-SH2 function. The study would be strengthened by evidence that MN714 binding to SOCS2 in cells (or MN551 in cell lysates) inhibits binding to phosphorylated proteins. Ideally, this would translate to a functional outcome; however, I'm cognisant that this is not easy to assay. The reported roles for SOCS2 include inhibition of growth hormone signalling and NFkB signalling, with the majority of published in vitro cellular assays showing only modest inhibition of signalling by SOCS2, without the robustness required to test inhibitor activity. Indeed, as pointed out by the authors, the binder described in this publication will be a welcome tool to investigate the functional role of SOCS2 in various systems.

If it can be measured, the affinity of MN551 for SOCS2 should be reported.

There is a typo in figure 2 legend title.

We thank the Editors and the Reviewers for their critical reading of our manuscript and for the insightful and constructive suggestions on how to improve the paper. We are encouraged by the overall positive feedback and have endeavoured to address their critique experimentally. Specifically, we have performed new experiments to deepen understanding the mechanism of action of our SOCS2 inhibitor. This new data, together with the already extensive structure-based design and structure-activity relationship data included at first submission, further establishes and qualifies MN551 (and its pro-drug MN714) as suitable chemical handle and probe for SOCS2. We have incorporated these new data into the revised manuscript:

1. Differential scanning fluorimetry (DSF):

- New DSF data evidencing MN551-induced stabilisation of SOCS2-EloBC (SBC2) (**new Figure 2D**).

2. Isothermal titration calorimetry (ITC):

- New ITC data showing titration of MN551 into SBC2 to establish thermodynamic binding parameters (**Figure 2E**).
- New ITC titrations of a) GHR_pY595 peptide into SBC; b) GHR_pY595 into SBC that was pre-incubated with one molar excess of MN551. Overlay of the data resulting from these two titrations clearly evidences that MN551 fully occupied SBC and irreversibly blocked its ability to bind the GHR_pY595 peptide (**new Figure 2F**).

3. Intact electrospray ionization mass spectrometry (ESI-MS):

- ESI-MS analysis of CISH-EloBC pre-incubated with MN551, evidencing covalent modification of CISH by the ligand (**new Figure 3D**). A structural superposition of the SH2 domains from SOCS2 and CISH, highlighting the conservation in the positioning of a Cys residue in both proteins, is included in **new Figure 3G**.

4. Pulldown of cellular SOCS2 with GHR_pY595 peptide and competition by MN551

- Pulldown of native SOCS2 from K562 cells using biotinylated GHR_pY595 peptide immobilized into agarose beads. Pulldown of SOCS2 was competitively blocked by pre-incubating cell lysates with MN551 in a concentration-dependent manner, as well as by pre-treating cells with MN714 (**new Figure 6A, 6B and 6C**)

Specific revisions in the figures include:

- **Figure 2** – additional panels 2D, 2E and 2F added to further characterise the binding of MN551 to SOCS2 protein *in vitro*.
- **Figure 3** – additional panel 3D to include the ESI-MS data of the covalent modification of CISH-EloBC by MN551, and panel 3G to show the structural superposition of the SH2 domains of SOCS2 and CISH.
- **Figure 4** – cLogP values added beneath the prodrug structure in panel 4C
- **Figure 6** – panels added to demonstrate cellular competition to GHR-SOCS2 binding by MN551 and MN714
- **Supplementary Figure 4** – panel C added to demonstrate calculation of IC₅₀ and K_i at time t=0

Below are point-by-point responses to each Referee's comments and when needed, reference to the corresponding changes in the manuscript made to address these. We believe the manuscript is now much improved from the review and resubmit for consideration at *Nature Communications*.

REVIEWER COMMENTS

Reviewer #1 (Remarks to the Author):

Ramachandran et al. present an exciting manuscript in which they developed a phosphotyrosine (pY)-masked covalent ligand that targets the SH2 domain of the SOCS2 E3 ligase. SOCS family proteins are central negative regulators of cytokine signaling pathways and some members are emerging targets for cancer immune therapies. SH2 domains are notoriously difficult to target and little progress to develop selective, potent, metabolically stable and cell permeable binders/inhibitors were made over the past two decades despite considerable efforts in big pharma, biotech and academia. In this paper, starting with masked pY ligand fragments and through a med chem tour-de-force, the authors develop SOCS2-targeting compounds with binding affinities in the hundred nM range. Co-crystal structures identify serendipitous covalent engagement of the lead compound (cpd 13) with a Cys in the EF loop of the SOCS2 SH2. Exchange to a more reactive covalent warhead resulted in MN551 that is subsequently characterized with great scientific rigour and depth (and at a high standard that is typically only found in preclinical drug discovery departments of major biotech/pharma companies). Cellular target engagement is demonstrated as well as unmasking of the pY group in cells.

Overall, this is an extremely exciting paper that convincingly reports an SH2 domain-targeting compound that covalently binds SOCS2 in cells and has high enough potency to serve as a SOCS2 inhibitor. In my opinion, this is the first report of an SH2 domain inhibitor that combines sufficient potency, cellular activity and high selectivity (through covalent engagement of target-selective cysteine). As discussed by the authors, this work has important implications for expand PROTAC/molecular glue degraders besides utility of the developed compound to study and perturb cytokine signaling pathways.

I already read a previous version of the manuscript in bioRxiv with great interest and excitement and was very happy when I was asked to review the manuscript for Nature Communications.

We thank the reviewer for a concise summary of our work and highlighting its novelty and significance to the field.

There are some minor points that I would like the authors to answer:

1. Selectivity within SOCS family: Selectivity for SOCS4 and SOCS6 is studied (Fig 3), but I wonder what the affinity for CISH is? The length of the EF loop in CISH and SOCS2 is identical, the sequence is conserved with two positive aa in the C-terminal part of the loop and the Cys is at the same relative position in the two protein. There is no CISH-EloB/C crystal structure published but the recombinant protein was produced previously. Please comment.

We appreciate the suggestion and agree with the reviewer that it was interesting to interrogate binding to CISH. To address this point, we have now expressed and purified CISH-EloBC complex and performed new ESI-MS binding experiments with it, and we include the new data in **new panel 3D to Figure 3**. The new data show the intact ESI-MS analysis of the CISH protein pre-incubated with MN551, evidencing covalent modification of CISH by MN551. To better understand this new data and allow prediction of the potential residue of CISH modified by our covalent compound, we have included a structural superposition of the SH2 domains of SOCS2 and CISH respectively (**new panel G in Figure 3**). CISH has the highest sequence identity to SOCS2 SH2 and, as the reviewer has indeed pointed out, contains a Cysteine residue (Cys144) in the EF loop at the same relative position of Cys111 in SOCS2. Therefore, the new data supports covalent modification of CISH at this conserved site by MN551.

To further discuss the SOCS-family wide selectivity of our inhibitor, we have made additions to the **'MN551 selectively reacts with SOCS2 and CISH over other SOCS family members'** subheading of the Results text:

“Beyond SOCS2, SOCS4, SOCS5, SOCS7 and CISH also have a cysteine in the same loop (Figure 3A), and given the expected conserved ligand binding mode, they could all be engaged by MN551. To verify our hypothesis, we incubated recombinant SBC complexes of SOCS4, SOCS6 and CISH with MN551 for 2 hours and analysed for covalent modification of the SOCS proteins by intact ESI-MS (Figure 3B-D). These proteins were chosen as SOCS4 and 6 are already structurally well characterized, and because SOCS4 Cys350, and CISH Cys144 are in a similar position as Cys111 in SOCS2, whereas SOCS6 is lacking cysteines in the EF loop37-39. We observed 18% modification of SOCS6 and complete modification of CISH. In contrast, MN551 failed to covalently engage SOCS4. An overlap of crystal structures of SOCS2 and SOCS4 show that although the Cys350 from SOCS4 is in the same loop as Cys111 (SOCS2), the MN551 binding mode and distance from the reactive group of MN551 may not be ideal for covalent bond formation (Figure 3E). Although SOCS6 lacks a cysteine in its EF loop, Cys471 from the BG loop is predicted to be in close proximity to the chloroacetamide functional group of MN551, as evident from the superposed SOCS2 and SOCS6 structures, potentially explaining the modification, albeit

minimal (Figure 3F). Structure-guided improvements upon this observed MN551-SOCS6 covalency could expedite future attempts in development of covalent binders for SOCS6. In the absence of a PDB structure for CISH, we overlaid SOCS2 SH2 domain with an AlphaFold model generated for the CISH SH2 domain with deletions in the N-terminus and the PEST sequence (66-258, Δ 174-202, Figure 3G). Amongst all the SOCSs, CISH has the highest sequence identity to SOCS2 SH2 and the presence of a conserved Cys144 in the EF loop together likely explain its complete covalent modification by MN551.”

2. Does MN551 act as a pY peptide competitor? Previous work of the Ciulli lab showed high-affinity binding and the structural basis for EpoR and GHR binding of SOCS2. Is MN551 competing with these pY peptides, e.g. in an FP assay in vitro?

Thank you for the suggestion, we agree this is a good point. We have now performed new competitive ITC experiments and include the results in **new Figure 2F**. The new data show pre-incubation of SBC with a molar excess of MN551 (1:2 of compound relative to SOCS2) completely blocks binding of GHR_pY595 to SBC. This observation demonstrates that MN551 irreversibly binds the SH2 domain of the SOCS2 in a manner that is, as expected, competitive with GHR_pY595 binding.

We have made additions to the ‘**MN551 is a Cys111-specific covalent SOCS2 ligand that blocks substrate binding**’ subheading of the Results text:

“To confirm that MN551 blocks binding of the natural substrates of SOCS2, we performed competition ITC experiments of GHR_pY595 peptide against SBC either unmodified or preincubated with equimolar amount of MN551 for 2 hours at room temperature. In the

presence of MN551, GHR was no longer able to bind to SBC, nor to compete out the inhibitor, demonstrating covalent saturation of the SOCS2 binding site by MN551 (Figure 2F)."

3. Although given the large amount of fantastic data, I feel slightly embarrassed to ask this: Do the authors have some preliminary data that they would want to include to show activity of MN551 in a cellular read-out, e.g. hyperactivation of a cytokine signaling dependent readout?

We agree that the ultimate goal of our research is to develop biologically active compounds that work in cells. As a series of major steps towards that goal, we describe here the discovery of the first small molecule inhibitor of the SOCS2-substrate interaction. In addition, as the first-in-class inhibitor of this interaction, we qualify the compounds as cell permeable via our pro-drug strategy and demonstrate covalent modification of SOCS2 and blockade of the SOCS2-substrate interaction inside cells. We have indeed begun to investigate potential cellular activities of our compounds. However, it is not straightforward nor established how to best assess functional activity of a SOCS2 inhibitor within cell. This point was also acknowledged by Reviewer 3, see below. Therefore, while we agree that demonstrating biological activity for these compounds is a desired long-term goal, we respectfully assert that those experiments lie outside of the scope of our current work and that the results presented here merit publication on their own.

Reviewer #2 (Remarks to the Author):

The manuscript by Ciulli and colleagues describes the discovery of a selective covalent inhibitors of the E3 ubiquitin ligase SOCS2. The work covered the design, synthesis, structural biology, metabolism and cell activity of the compounds. Overall, this is a rather comprehensive manuscript that builds on recent progress on the discovery and cell delivery of pY containing drugs. The use of pY prodrug strategies to improve the cellular uptake of the compounds is notable. As mentioned above this is a rather comprehensive manuscript, but few comments for the authors to consider.

We thank the reviewer for their excitement about the manuscript and about its comprehensive and wide-ranging nature.

- The authors summarised the functions and signalling of SOCS proteins well. However, many will find this rather detailed and thus the authors should include a Figure showing JAK/STAT signalling and SOCS.

We thank the reviewer for their comment. We agree that it is important to introduce the function and role of SOCS2 protein, both in cell signalling and as component of a Cul5 ligase complex. However, respectfully, we do not feel that a figure is required to show the biology, because this is well covered in several review articles (see PMID: 34604264; PMID: 23545160). Furthermore, considering the number of figures we already include in the paper we feel it best not to add further to those.

- Again, the text description of the SOCS2 structure binding to the phosphopeptides was very good, but a Figure showing this would make it easier to understand and visualise the key interactions with pY.

Thank you for the suggestion. We already illustrate the key interactions mediated by the phosphate group of pY and pY-containing peptides and compounds with the SOCS2 SH2 domain right upfront in Figure 1A and 1B.

- The authors need to include the calculated logP values of the compounds and integrate these in the discussion in terms of cell permeability. It is common that the prodrugs can be associated with high lipophilicity, which limits their water (cell media) solubility when treating cells.

We appreciate the comment and have added the cLogP values to Fig 4C and included the following text to the manuscript:

“Furthermore, the increased hydrophilicity of POM protected MN714 with respect to the aryloxy phosphoramidate prodrugs likely enhanced the cell-permeability. MN714 (cLogP 4.329) was significantly less lipophilic than all phosphoramidate prodrugs tested (cLogP ranging from 4.729 to 5.818), placing it firmly within the desired range for lipophilicity as outlined in Lipinski’s rule of 5.”

- Page 7, lines 234-235, sentence “A co-crystal structure of SBC....of MN551 and cyc111” needs to be followed by the Figure number referring to the co-crystal structure and mass spectrometry data.

Thank you for highlighting the oversight. We have now added the reference to the figures.

- In the discussion around the pY prodrugs, the authors need indicate that the aryloxy triester phosphoramidate prodrugs are generally poorly metabolised in HeLa cells due to the low expression of the enzymes that metabolize these prodrugs in HeLa cells. See Miccolli et al <https://doi.org/10.1021/acs.jmedchem.1c01490> who examined the enzymes expression across a selection of cells including HeLa cells.

Thank you for bringing this to our attention. We have now included this reference in the discussion as a potential reason for low potency of compounds **21**, **22** and **23**. We have also amended the text in the Discussion:

“The low efficacy of aryloxy triester phosphoramidate prodrugs can be explained by the lower levels of expression of the enzymes that unmask these prodrugs in HeLa cells.”

- Also, in describing the pY prodrugs data, the authors need to comment on the incubation time being 20h, which is suitable for POMtide prodrugs, but the aryloxy triester prodrugs are often incubated for longer times > 24 h to see some effect.

The assay was performed for 20 hours with an intention to rank the prodrugs for their permeability and deprotection. We refrained to incubate for longer time points to minimize any risk of cellular toxicity. We have also amended the text in the Discussion:

“The low efficacy of aryloxy triester phosphoramidate prodrugs can be explained by the lower levels of expression of the enzymes that unmask these prodrugs in HeLa cells.”

- Although in the discussion the authors focused on the potential future use of MN551/714 in the design of PROTACs, an equal emphasis and future outlook on the use of MN551/714 alone

to modify SOCS2 (JAK/STAT signalling) is needed since covalent inhibitors are being widely pursued in drug discovery.

Thank you very much for the suggestion. We have now amended the Discussion to highlight the significance of the MN551/MN714 as a covalent inhibitor

“The chemical probes could provide long awaited insight into the role of SOCS2 in regulation of growth hormone signalling and NF- κ B signalling. Future work will also focus on establish the functional impact of cellular inhibition of SOCS2, and the intra-SOCS family selectivity inhibition, thereby demonstrating full application of MN551/MN714 and/or their non-covalent analogues as probes to study the JAK/STAT signalling pathway. The efficient covalent modification of recombinant CISH (66-258, Δ 174-202) by MN551 justifies future biophysical, structural, and cellular characterisation to verify if the covalency translates to full length intracellular CISH. Considering the interest in inhibition of CISH as an antitumor therapeutic strategy aimed at enhancing activity of Natural Killer (NK) cells and decreasing PD-1 expression levels, chemical probes targeting selectively CISH would be of great utility.”

- Finally, the figures resolution is poor; authors needs to have high resolution figures.

Thank you for pointing out the issue. Source Illustrator images are now included with the revised submission of the manuscript which will ensure high resolution in the final reproduction.

Together, this manuscript and after the corrections will be provide a useful blue print for the future discovery of SH2 domain inhibitors.

We fully concur with the reviewer, and thank you for the vote of support!

Reviewer #3 (Remarks to the Author):

The manuscript by Ramachandran and colleagues, describes the chemical development of a small chemical binder of the SOCS2-SH2 domain.

SH2 domains have traditionally been difficult to drug, based on the common mode of binding across the SH2 family, and the requirement for a negatively charged phospho-tyrosine that impedes membrane permeability. In addition, once inside the cell, there is the potential for de-phosphorylation. The authors overcame these constraints by generating a covalent pro-drug based on pTyr.

This study represents a significant advance in the field, given the challenges in targeting SH2 domains, and the limited number of binders described to date that target the SH2-pTyr binding pocket. This is the first description of a binder targeting the pTyr pocket in the SOCS family proteins.

This is a well-written, comprehensive and informative study. However, it is not clear whether the prodrug MN714 will act as an inhibitor of SOCS2-SH2 function. The study would be strengthened by evidence that MN714 binding to SOCS2 in cells (or MN551 in cell lysates) inhibits binding to phosphorylated proteins. Ideally, this would translate to a functional outcome; however, I'm cognisant that this is not easy to assay. The reported roles for SOCS2 include inhibition of growth hormone signalling and NF κ B signalling, with the majority of

published in vitro cellular assays showing only modest inhibition of signalling by SOCS2, without the robustness required to test inhibitor activity. Indeed, as pointed out by the authors, the binder described in this publication will be a welcome tool to investigate the functional role of SOCS2 in various systems.

We thank the reviewer for the concise summary highlighting the significance of the work. We agree that it is important to demonstrate the activity of MN714 in cellular context. We are indeed looking into identifying biomarkers that would help establish the functional utility of MN551/MN714 as tool compounds in understanding SOCS biology. In the meantime, we took on board the suggestion and to address this we performed new peptide-pull-down experiments. We immobilized biotinylated GHR_pY595 peptide onto streptavidin-agarose beads to pull-down endogenous SOCS2 from K562 lysates. We performed a dose-dependent competition experiment of MN551 with biotinylated-GHR_pY595 recovery of endogenous SOCS2 through pull-down from K562 lysates. MN551 clearly demonstrates blockade of SOCS2 pull-down in a dose-dependent manner, with an apparent IC₅₀ of 9 μM (**new Figure 6A, 6B**). Furthermore, we compared peptide pull-down of endogenous SOCS2 from K562 cells pre-treated with either vehicle DMSO or with 10 μM MN714 for 6 hours. The data clearly shows that treatment with MN714 blocked pull-down of SOCS2 (**new Figure 6C**). Together these pull-down experiments evidence that MN551/MN714 can competitively block recruitment of SOCS2 protein to its native substrates within cells.

If it can be measured, the affinity of MN551 for SOCS2 should be reported.

We have now performed a new ITC titration of MN551 into SOCS2-BC and report the K_d and other thermodynamic binding parameters measured from this experiment in **new Figure 2E**.

E
Also, in the supplementary material (new Supp Figure 4C) we now report the calculated K_i at $t=0$ (determined by extrapolating IC_{50} at $t=0$). This should give reversible binding affinity of MN551.

C

There is a typo in figure 2 legend title.

Thank you for highlighting the mistake. The legend is now corrected.